# Comparative analysis of the myoglobin gene in whales and humans reveals evolutionary changes in regulatory elements and expression levels

Charles Sackerson[1]*, Vivian Garcia[1¤a], Nicole Medina[1¤b], Jessica Maldonado[1¤c], John Daly[1], Rachel Cartwright[1,2]

**1** Biology Department, California State University Channel Islands, Camarillo, California, United States of America, **2** The Keiki Kohola Project, Lahaina, Hawaii, United States of America

¤a Current address: Department of Stem Cell and Regenerative Biology, Harvard University, Cambridge, Massachusetts, United States of America
¤b Current address: Agilent Technologies, Carpinteria, California, United States of America
¤c Current address: Ventura County Sheriff's Office Forensic Services Bureau, Ventura, California, United States of America
* charles.sackerson@csuci.edu

**Data Availability Statement:** All relevant data are within the paper and its Supporting Information files.

## Abstract

Cetacea and other diving mammals have undergone numerous adaptations to their aquatic environment, among them high levels of the oxygen-carrying intracellular hemoprotein myoglobin in skeletal muscles. Hypotheses regarding the mechanisms leading to these high myoglobin levels often invoke the induction of gene expression by exercise, hypoxia, and other physiological gene regulatory pathways. Here we explore an alternative hypothesis: that cetacean myoglobin genes have evolved high levels of transcription driven by the intrinsic developmental mechanisms that drive muscle cell differentiation. We have used luciferase assays in differentiated C2C12 cells to test this hypothesis. Contrary to our hypothesis, we find that the myoglobin gene from the minke whale, *Balaenoptera acutorostrata*, shows a low level of expression, only about 8% that of humans. This low expression level is broadly shared among cetaceans and artiodactylans. Previous work on regulation of the human gene has identified a core muscle-specific enhancer comprised of two regions, the "AT element" and a C-rich sequence 5' of the AT element termed the "CCAC-box". Analysis of the minke whale gene supports the importance of the AT element, but the minke whale CCAC-box ortholog has little effect. Instead, critical positive input has been identified in a G-rich region 3' of the AT element. Also, a conserved E-box in exon 1 positively affects expression, despite having been assigned a repressive role in the human gene. Last, a novel region 5' of the core enhancer has been identified, which we hypothesize may function as a boundary element. These results illustrate regulatory flexibility during evolution. We discuss the possibility that low transcription levels are actually beneficial, and that evolution of the myoglobin protein toward enhanced stability is a critical factor in the accumulation of high myoglobin levels in adult cetacean muscle tissue.

**Funding:** This work was supported by internal grants from California State University Channel Islands to CS: Research, Scholarship and Creative Activity Grant #811-GD970 (2016), Faculty Research and Development Minigrant (2016), Student Research Steering Council Microgrant (2015), Provost's Faculty Resource Fund Grant #GD945 (2011). Additional support came from instructional funding for Independent Research (Bio 494). The funders had no role in study design, data collection and analysis, decision to publish, or preparation of the manuscript.

**Competing interests:** The authors have declared that no competing interests exist.

## Introduction

The Cetacea diverged from their terrestrial relatives about 50 million years ago [1–3] and have since undergone numerous adaptations to their aquatic lifestyle. Among these is high levels of the intracellular hemoprotein myoglobin (MB) in skeletal muscle [4], especially in those muscles required for swimming. Myoglobin stores oxygen and assists in its facilitated diffusion to supply oxygen in muscle tissue during extended dives [5]. Myoglobin also manages reactive oxygen and nitrogen species [6], regulates mitochondrial respiration [7], and carries out other functions relevant to a relatively anoxic muscle environment [8]. These functions all contribute to muscle fitness during exercise in an oxygen limited environment.

We are interested in the mechanisms underlying the evolution of the high levels of myoglobin in cetacean muscles. Traditionally, the study of evolutionary change has focused on changes in amino acid sequence, but recent studies have highlighted the importance of changes in gene regulation as drivers of evolutionary change [9–11]. Therefore, we hypothesized that the high myoglobin protein levels may be the result of evolutionary adaptations affecting the regulation of the myoglobin gene, which lead to high rates of transcription and, hence, protein. That is, even in the absence of induction by exercise or other physiological gene regulatory influences, the constitutive, or basal, level of transcription would be sufficient to lead to high myoglobin protein levels. Thus, the steady accumulation of intracellular myoglobin during the animal's lifespan [12] would be driven by these constitutive mechanisms.

Whereas the regulatory sequences that control muscle-specific expression of the myoglobin gene have been well studied in humans and mice [13–16, reviewed in 17], little is known about the regulation of cetacean myoglobin genes. A summary of the 5' regulatory region of the human gene is shown in Fig 1. Of special note is a muscle-specific core enhancer identified in humans that responds to the developmental signals that trigger muscle cell differentiation. The two components of this enhancer, the CCAC-box and the AT element, act synergistically to drive high levels of transcription [15, 18]. Mutations in either reduce transcription to 10–20% of wild-type [14]. Based on these findings, we cloned the 5' regulatory region from several

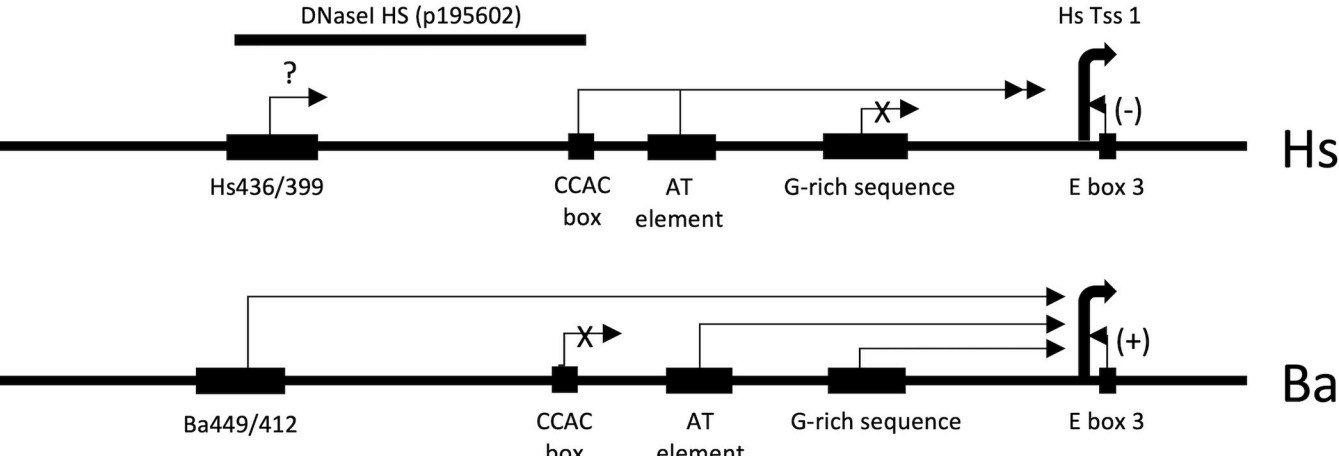

**Fig 1. Summary of regulatory features in the myoglobin 5' flanking region.** DNA sequence elements identified previously or in this work are schematized, to scale. The top line represents the human (*Homo sapiens*, Hs) gene, the lower line the minke whale (*Balaenoptera acutorostrata*, Ba) gene. Hs Tss1 is the human major transcription start site (Tss) [19]; the Ba Tss is presumed to be at the same nucleotide. The arrows represent positive regulatory inputs unless otherwise indicated. The "+" and "-" notation on the E-box3 arrows reflect its activating (positive) effect in the Ba gene and repressive (negative) effect in the Hs gene. The "X" on the Hs G-rich sequence arrow and the Ba CCAC-box arrow indicate a lack of effect. The "?" on the conserved Hs orthologue of the Ba449/412 region indicates that its effect on expression was not tested. DNaseI HS (p195602) indicates the extent of a DNaseI hypersensitive site identified in muscle cells by the ENCODE project as displayed in the UCSC Genome Browser [20, 21] (see Fig 7 for further details).

species spanning the evolutionary range from cetaceans to humans and assayed their transcriptional activity in the mouse myoblast cell line, C2C12, after differentiation to myotubes.

To further understand the transcription of the cetacean myoglobin gene, we then carried out a detailed dissection of the 5' flanking region of the minke whale (*B. acutorostrata*, Ba) gene, as a model for cetaceans in general. These results are summarized in Fig 1.

We report here two major findings from these studies. First, contrary to our hypothesis, we find that cetacean myoglobin genes do not have high transcriptional activity compared to the human gene; for example, the minke whale gene has an expression level of only ~8% that of humans. This implies that additional mechanisms are important to explaining the high myoglobin levels. These may include the induction of increased transcription in response to physiological signaling pathways [12, 17, 22, 23], and evolution of the MB protein toward increased stability and solubility [24, 25]. Second, considerable regulatory evolution has occurred in the myoglobin gene since a common ancestor with humans. Some regulatory elements such as the AT element are conserved in function, others such as the CCAC box have lost function, and novel regulatory elements are found that contribute to transcriptional activity. Cumulatively, these studies point to diverse mechanisms through which evolution can occur while satisfying a necessary goal: high levels of muscle-specific expression of the myoglobin protein.

## Results

### Species survey

Previous studies of human (*Homo sapiens*, Hs) myoglobin (MB) regulation have indicated that the important regulatory inputs are present in about 700 nucleotides (nt) of 5' flanking sequence [17, 26]. We cloned and sequenced about 700 nt of 5' flanking sequence from the MB genes of the baleen whales, minke whale, *B. acutorostrata* (Ba710, 710 nt of DNA 5' of the translational start codon (ATG) from Ba), and gray whale, *Eschrichtius robustus* (Er701). Alignments with the published human sequence showed that these whale regions corresponded to 671 nt of Hs sequence; we therefore also cloned 671 nt of human sequence (Hs671) for comparative studies. These promoter regions were placed 5' of the luciferase reporter in pGL4.10[luc2], transfected into the mouse myoblast cell line C2C12, and luciferase activity was measured after 4–6 days of differentiation to myotubes (see S1A File and Materials and Methods for details). We find that the whale promoters have relatively low activity compared to the human promoter, with Hs671 being >12-fold more active than Ba710 (Table 1, S2 File, Fig 2).

Mean activity of selected species, expressed as firefly luciferase (F) counts normalized first to a cotransfected renilla luciferase (R) internal standard, then secondly to a minke whale (Ba710) control included in duplicate in each transfection (F/R/Ba). The number of individual transfections, done in duplicate, is indicated as "n"; for example, n = 5 refers to 10 transfected wells in 5 independent experiments (see S10 File). pGL4.10 is empty vector. The Ba710 sample shown reflects independent transfections of Ba710, treated the same as the other samples. pGL4.10: empty vector, Ba710: *Balaenoptera acutorostrata*, Er701: *Eschrichtius robustus*,

**Table 1. Species scan of MB promoter activity.**

| Clone: | pGL4.10 (vector) | Ba710 (minke) | Er701 (gray) | Dc706 (dolphin) | Pp702 (porpoise) | Bt695 (cow) | Cc696 (elk) | Ec675 (horse) | Cf708 (dog) | Hs671 (human) |
|---|---|---|---|---|---|---|---|---|---|---|
| Mean (F/R/Ba) | 0.03 | 1.04 | 0.79 | 0.86 | 0.61 | 0.88 | 0.72 | 74.20 | 1.59 | 12.75 |
| n | 5 | 5 | 4 | 4 | 6 | 5 | 5 | 4 | 4 | 6 |

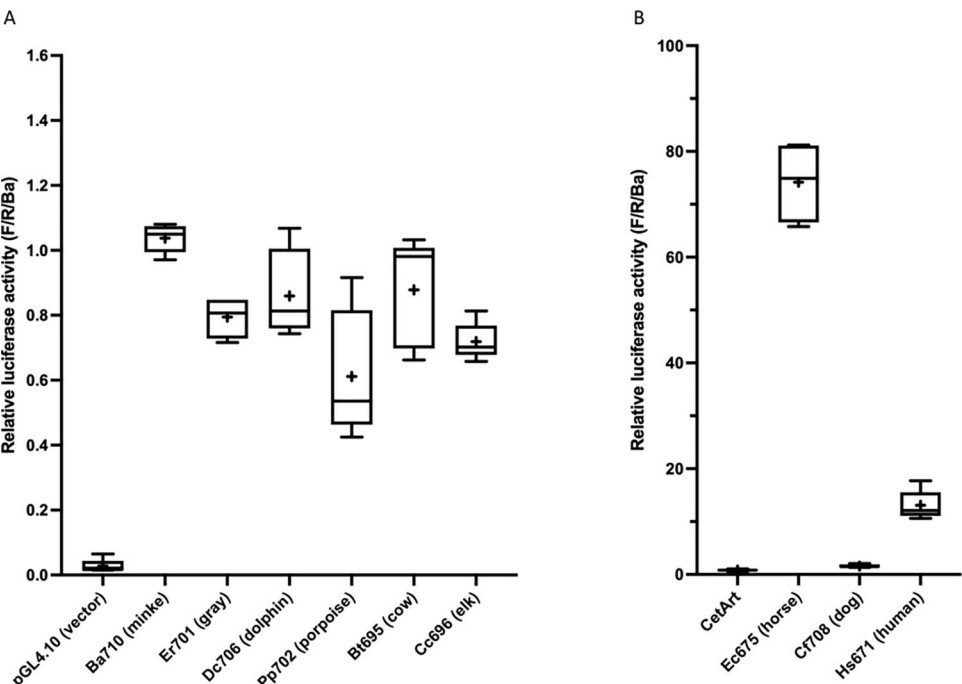

**Fig 2. Species scan of MB promoter activity.** (A) Box-and-whisker plots of selected cetacean and artiodactylan species, normalized as described for Table 1. The Y-axis shows activity compared to the Ba710 control included in each transfected plate (relative activity of the Ba710 control = 1.0). The "+" in each box is the sample mean. Clone designations are as described for Table 1. (B) The mean of ten cetacean and artiodactylan species ("CetArt", S2 File) is compared to horses (Ec675: *E. caballus*), dogs (Cf708: *C. familiaris*), and humans (Hs671: *H. sapiens*). Note that the axes differ in scale in A and B.

Dc706: *Delphinus capensis*, Pp702: *Phocoena phocoena*, Bt695: *Bos taurus*, Cc696: *Cervus canadensis*, Ec675: *Equus caballus*, Cf708: *Canis familiaris*, Hs671: *Homo sapiens*. Numbers indicate the size of each cloned promoter in base pairs, where the A of the translational initiating ATG = +1, and counting in the 5' direction begins with the nucleotide immediately 5' of the ATG, invariably a C in these species; thus, Ba710 has 710 nucleotides 5' of the ATG. The transcription start site (Tss) is not used as +1 because numerous human Tss have been identified in different sources, and the Tss from the other species has not always been experimentally determined.

We pursued these results by cloning and testing comparable regions from the toothed whales, common dolphin, *Delphinus capensis* (Dc706), and harbor porpoise, *Phocoena phocoen*a (Pp702); the closely-related terrestrial artiodactylan (even-toed ungulate) species, cows, *Bos taurus* (Bt695), and elk, *Cervus canadensis* (Cc696); and more distantly-related species, horses, *Equus caballus* (Ec675, an odd-toed ungulate), and dogs, *Canis familiaris* (Cf708, a carnivore). Toothed whales, artiodactylans, and dogs are not greatly different in activity from baleen whales (Table 1, S2 File). The promoter from horses is highly active, >5-fold more so than humans, and >70-fold that of the minke whale (Table 1, S2 File). This species survey is summarized in Fig 2.

This survey demonstrates that the high levels of myoglobin protein in cetacean muscle is unlikely to be due to a high constitutive level of transcriptional activity of the MB gene. We use the term "constitutive" to describe expression driven by differentiation of the C2C12 cells. Our assays do not address increases in expression "induced" by physiological signals such as calcium flux, hypoxia, or lipid availability.

## Exploration of the minke whale (*Ba*) myoglobin gene regulatory regions previously identified in the human (*Hs*) gene

The low activity of the cetacean genes prompted us to explore in detail the transcriptional regulation of a model cetacean MB gene, that of the minke whale, *B. acutorostrata* (Ba). Previous work on regulation of the Hs myoglobin gene identified a muscle-specific core enhancer composed of two sub-regions: an "AT element" and a "CCAC-box" [13–15, 17]. In addition, an isolated E-box, "E-box3" [27] was identified in exon 1. The orthologous sequences from the Ba gene were identified by DNA sequence alignment and assessed for their contributions to gene regulation in differentiated C2C12 mouse myoblast cells. Binding sites for the transcription factor NFAT [28] are also able to be identified by sequence conservation or transcription factor searches, but our assays do not include manipulations that would activate the NFAT pathway, so these binding sites were not explicitly pursued.

**AT element.**   The AT element consists of three separate transcription factor binding regions: an AT-rich region flanked on both sides by E-box sequences (Fig 3A). The AT-rich region has been shown to bind a critical activating transcription factor, MEF2 [18]. E-box sequences bind muscle-specific basic helix-loop-helix (bHLH) transcription factors such as MYOD and myogenin (MYOG) in cooperation with the ubiquitous TCF3/E2A proteins, E12 and E47 [29]. The AT element is highly conserved across the species studied (S3A File). There are two nucleotide differences between Hs and Ba affecting the MEF2 binding consensus [30] (Hs: CTAAAATAG → Ba: TCAAAATAG), but the Ba AT element is still predicted (LASAGNA, rVISTA; see Materials and Methods) to bind MEF2. The two E-boxes are perfectly conserved in the species examined, except E-box1 in porpoises (see S3A File).

In all, three nucleotides differ between Ba and Hs within the AT element (Fig 3A). When these three nucleotides were mutated in the Ba promoter to match the Hs sequence (AT swap) the resulting activity was unchanged relative to the intact Ba710 control (Fig 3B).

Previously, mutation of the AT-rich core of the MEF2 site in the Hs promoter ("MEF2 mut") was shown to reduce expression to about 20% of wild-type [14]. To test the MEF2 site in the Ba promoter for function, we reproduced this mutation in the Ba promoter to create Ba MEF mut (Fig 3A). In Ba, this mutation reduced expression only slightly, to 88% of control (Fig 3B), not a statistically significant reduction. Mutations in the two E-boxes (Ba E-box1 mut and Ba E-box2 mut, Fig 3A) also reduced activity only slightly, to 90% and 91% of control, respectively (Fig 3B). In contrast, when we deleted the entire AT element (Ba ΔAT, deletion (Δ) of Ba248/222 and replacement with a GAATTC sequence, Fig 3A), expression is reduced to 68% of control (Fig 3B). When multiple comparisons were made between AT swap, Ba MEF mut, Ba E-box1 mut, Ba E-box2 mut, and Ba ΔAT, the only mutation with a statistically significant difference from the others was Ba ΔAT (Fig 3B, S3D File). This indicates the MEF2 site does not play the dominant role in the Ba gene that it does in the Hs gene, and each of the three components of the Ba AT element are dispensable, such that mutation of any individual component has a minimal effect, but deletion of all three reduces expression by more than 30%.

**CCAC-box.**   The CCAC-box is a GC-rich region 5' of the AT-element, first defined as a "myoglobin upstream regulatory element" (MbURE) critical for transcriptional activity [13]. The CCAC-box is reported to bind the ubiquitous transcriptional activator SP1 [16] but may also bind other transcription factors [14, 17, 31] (S4A File). In the Hs gene, the AT element and CCAC-box interact synergistically to drive muscle-specific expression [15, 18].

Mutation of the CCAC-box in the Hs gene reduces activity to about 10% of wild-type [14]. However, the sequences around the CCAC-box are poorly conserved across species (S4A File). This raises the question of whether the Ba CCAC-box is a fully functional ortholog of the Hs

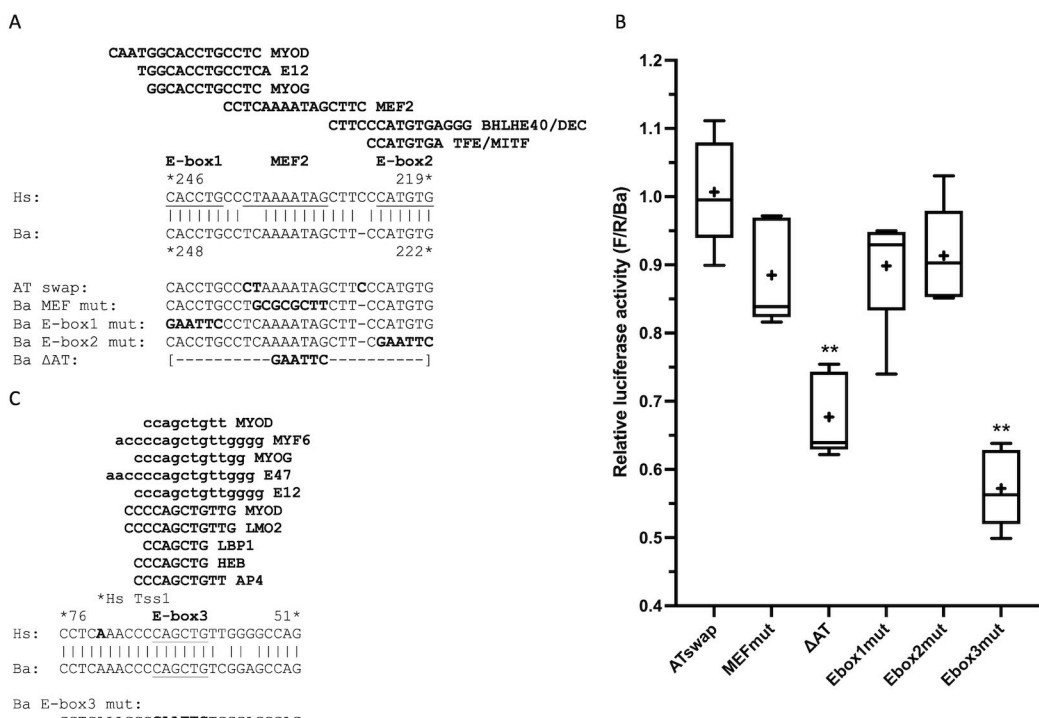

**Fig 3. Deletion of the entire AT element is required to impact Ba promoter activity.** (A) Alignment of AT element sequences from Hs and Ba, with predicted transcription factor binding sites conserved in both species (rVISTA) and expressed in muscle (S7 File) shown above; the human binding site sequences are shown here and below for simplicity. The two E-boxes and the core of the MEF2 binding site [30] are underlined. Below are the mutations tested. rVISTA predicts that Ba MEF mut eliminates MEF2 binding but does not impact binding of other factors; Ba E-box1 mut eliminates MYOD, E12, and MYOG binding but has no impact on MEF2 or E-box2 binding; Ba E-box2 mut eliminates DEC and TFE binding without affecting E-box1 or MEF2 binding. Ba ΔAT is not predicted (rVISTA) to bind any of the transcription factors shown. (B) Activity of AT element mutations. To determine which samples differ from control, ATswap was used for the comparisons (relative activity of the Ba710 control = 1.0); samples indicated by ** had p <0.01 (see S3D File for details). ATswap: AT swap, mean = 101% of control. MEFmut: Ba MEF mut, mean = 88% of control. ΔAT: Ba ΔAT, mean = 68% of control, p <0.0001. Ebox1mut: Ba E-box1 mut, mean = 90% of control. Ebox2mut: Ba E-box2 mut, mean = 91% of control. Ebox3mut: Ba E-box3 mut, mean = 57% of control, p <0.0001. (C) Alignment of the E-box3 region from Hs and Ba (E-box3 is addressed in the text below), with predicted transcription factor binding sites expressed in muscle (S7 File) shown above. Binding site sequences in lower case were identified independently on the Hs and Ba sequences by LASAGNA; binding sites in upper case were identified as conserved in Hs and Ba by rVISTA. MYOD binding is also predicted by MATCH. The T → C difference at nt 59 does not prevent binding for any of the transcription factors shown (numbering from the ATG is the same in Hs and Ba for this region). The human major transcription start site is indicated at nt 72. Note that all three E-boxes received the same mutation: CAnnTG → GAATTC.

CCAC-box. To test this, we reproduced the "CCAC mut 3" of Bassel-Duby et al. [14] in the Ba promoter, creating Ba CCAC mut (Fig 4A). We find that Ba CCACmut reduces activity to 85% of control (Fig 4B), not a statistically significant reduction (S4C File).

We next made a 67 nt deletion removing the CCAC-box and a surrounding C-rich stretch containing multiple CAC motifs (Ba ΔCCAC, replacement of Ba317/251 with a GAATTC sequence; Fig 4A). Ba ΔCCAC was found to increase activity to 110% of control (Fig 4B) rather than decreasing it as expected. We noted a conserved predicted SP1 site (rVISTA) immediately 5' of this deletion (Fig 4A) and hypothesized that the increased activity may be due to bringing this SP1 site closer to the AT element, and this may mask a possible loss of activity. However, extending the deletion to remove the SP1 site (Ba ΔSP1-CCAC, Ba334/251) still only slightly reduced expression, to 92% of control (Fig 4B).

In the Hs gene, the CCAC-box and AT element activate expression synergistically [15, 18]. To test whether combining Ba ΔCCAC with Ba ΔAT would display a synergy that would reveal

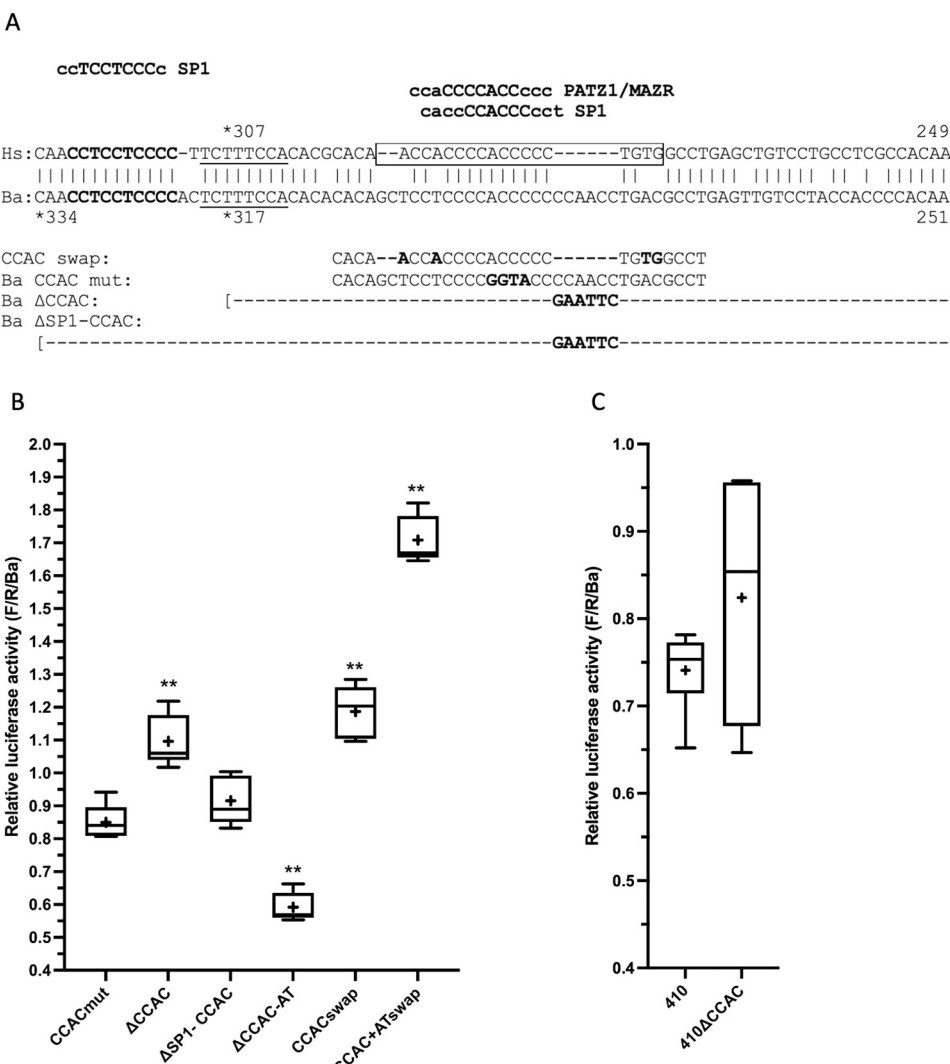

**Fig 4. The Ba CCAC-box has little detectable activity in differentiated C2C12 cells.** (A) Alignment of the CCAC-box region from Hs and Ba, with predicted transcription factor binding sites conserved in both species (rVISTA) and expressed in muscle (S7 File) shown above. An NFAT site not found by rVISTA but previously identified [32] is underlined. The 5' SP1 site targeted by Ba ΔSP1-CCAC is in bold. The region targeted by the Ba CCAC swap is boxed in the Hs sequence. Below are the mutations tested. rVISTA predicts that Ba CCAC mut eliminates the PATZ1/MAZR and SP1 sites but a new nonconserved SP1 site is predicted in the Ba CCAC swap (rVISTA: agctCCTCCCcgg); similarly, a new site is predicted in the Hs CCAC mut 3 [14] (rVISTA: acaaCCACCccgg/). (B) Activity of CCAC-box mutations. For analysis of statistical significance in this figure, Ba ΔSP1-CCAC was used for comparison; samples indicated by ** had p <0.01 (see S4C File for details). CCAC mut: Ba CCAC mut, mean = 85% of control. ΔCCAC: Ba ΔCCAC, mean = 110% of control, p = 0.001. ΔSP1-CCAC: Ba ΔSP1-CCAC, mean = 92% of control. ΔCCAC-AT: Ba ΔCCAC-AT, mean = 59% of control, p <0.0001. CCACswap: CCAC swap, mean = 119% of control, p <0.0001. CCAC+ATswap: CCAC+AT swap, mean = 171% of control, p <0.0001. CCACswap vs. CCAC+AT swap: p <0.0001. (C) Comparison of Ba410 to Ba410ΔCCAC. 410: Ba410, mean = 74% of control. 410ΔCCAC: Ba410 ΔCCAC, mean = 82% of control.

an effect of the Ba CCAC-box, we made Ba ΔCCAC-AT (contiguous deletion of Ba317/222, S1B File), which encompasses both regions. This larger deletion of 96 nt still did not show a statistically significant reduction in activity when compared to the Ba ΔAT deletion alone (p = 0.128, S4C File).

We also tested whether introducing the Hs CCAC-box and AT element sequences together into the Ba promoter would be sufficient to confer the high level of expression characteristic of

the Hs promoter. We mutated the Ba CCAC-box to match the Hs sequence (CCAC swap, Fig 4A); these changes increased activity to 119% of control (Fig 4B). When the nucleotide changes of the CCAC swap were combined with the three nucleotide changes of the AT swap (CCAC +AT swap), the activity increased synergistically to 171% of control.

Much of the previous work on the Hs MB gene, including the construction of CCAC mut 3 [14], used a clone ending at a HindIII site at Hs444 (corresponding to Ba 457). We hypothesized that redundant or compensating sequences present between Ba457 and Ba710 may be masking the influence of the Ba CCAC box mutations. We therefore tested the Ba ΔCCAC mutation in the context of a Ba promoter truncated to Ba410 (Ba410 ΔCCAC, deletion of Ba317/251, Fig 4C). The Ba410 endpoint leads to a reduction in expression to 74% of control; Ba410 ΔCCAC is expressed at 82% of control (S4D File), not a statistically significant difference by two-tailed t-test (Ba410 ($M = 0.741$, $SD = 0.046$), Ba410ΔCCAC ($M = 0.824$, $SD = 0.142$); $p = 0.269$).

Taken together, these data indicate that the Ba CCAC-box is not a major contributor to the activity of the Ba gene, as it is in the human gene. Deleting 84 nt from the CCAC-box region (Ba ΔSP1-CCAC) has little effect on expression. In contrast, introducing the human CCAC-box sequences into the Ba gene led to a statistically significant increase in expression, and the human CCAC-box interacts synergistically with the human AT element within the context of the Ba gene.

**AT element and CCAC-box deletions in the human gene.**   The above experiments prompted us to validate the ability of our assay system to detect deletions of the AT element and CCAC-box, so we made corresponding deletions in the Hs671 promoter. We find that deletion of the Hs AT element (Hs ΔAT, replacement of Hs246/219 with GAATTC, Fig 3A) reduced expression to 24% of the full length Hs671 promoter (Fig 5). This is similar in magnitude to the human MEF2 mut [14], but greater than the reduction to 68% seen with Ba ΔAT. Deletion of Hs CCAC (Hs ΔCCAC, replacement of Hs307/249 with GAATTC, Fig 4A) reduced expression to 65% of Hs671 (Fig 5). Hs ΔCCAC has a much more modest effect than the reduction to ~10% seen for the human CCAC mut 3 [14], but still greater than seen in any of our Ba CCAC-box mutations. Therefore, we conclude that the C2C12 assay system is capable of showing significant reductions in activity in response to the ΔAT and ΔCCAC deletions, and that the Ba CCAC-box and AT elements do not share the levels of activity previously shown in the Hs gene.

**E-box3.**   In addition to the two E-box sequences in the AT element, a third conserved E-box exists, in the 5'UTR of MB exon 1 (Fig 1). This E-box, termed E-box3 [27], is predicted (rVISTA, LASAGNA) to bind several transcription factors expressed in muscle (Fig 3C). E-box3 was explored in mice [27] and found to repress expression in a muscle-specific fashion, that is, its mutation increased expression. In contrast, we find that the same mutation used for E-box1 and E-box2 decreased activity to 57% of control (E-box3 mut, Fig 3B and 3C). Therefore, Ba E-box3 is an activating element, in contrast to its action as a repressive element in the mouse gene.

## Two novel regulatory regions are revealed in studies of the Ba gene

We used bioinformatics to guide further analysis of the Ba MB gene, and uncovered two additional functional regions, a G-rich sequence at Ba179/155, and a conserved sequence at Ba449/412.

**G-rich sequence.**   A search conducted for transcription factor binding sites (rVISTA) revealed that SP1 binding sites occur in only two places in the Ba gene: in the vicinity of the CCAC-box (Fig 4A) and in a strikingly G-rich sequence 3' of the AT element (80% G over 25

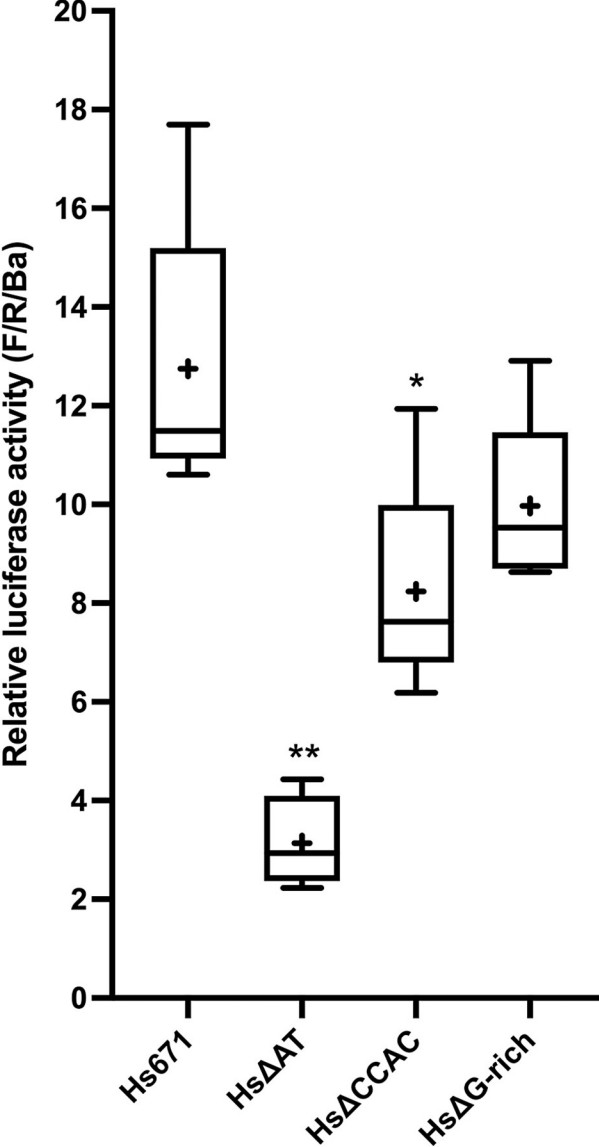

**Fig 5. The human AT element and CCAC-box have detectable activity in differentiated C2C12 cells.** Activity of deletions in Hs671, normalized to Ba710 (F/R/Ba). Samples that differ from Hs671 are indicated by asterisks (* p <0.05, ** p < 0.01; see S5B File for details). Hs671: mean = 12.7-fold Ba710. Hs ΔAT: mean = 3.1-fold Ba710 (24% of Hs671), probability of a difference from Hs671 (p) = <0.0001. Hs ΔCCAC: mean = 8.2-fold Ba710 (65% of Hs671), p = 0.019. Hs ΔG-rich (the G-rich sequence is addressed in the text below): mean = 10.0-fold Ba710 (79% of Hs671), probability of a difference from Hs671 is not significant.

nucleotides, Ba179/155, Fig 6A). The G-rich sequence is flanked at its 3' side by a repeated GAGA motif [33]. We therefore targeted this region for further study. A deletion of 25 nucleotides in the G-rich sequence (Ba ΔG-rich, Fig 6A) strongly reduced expression to 42% of control (ΔG-rich, Fig 6C). Deletion of the GAGA motif (Ba ΔGAGA, deletion of Ba155/145) had no effect (S6A File).

In the Hs promoter, the CCAC-box is proposed to bind SP1, which then interacts synergistically with transcription factors bound to the AT element [16, 18]. The relative inactivity of the Ba CCAC-box led us to hypothesize that this G-rich cluster of predicted SP1 sites may act

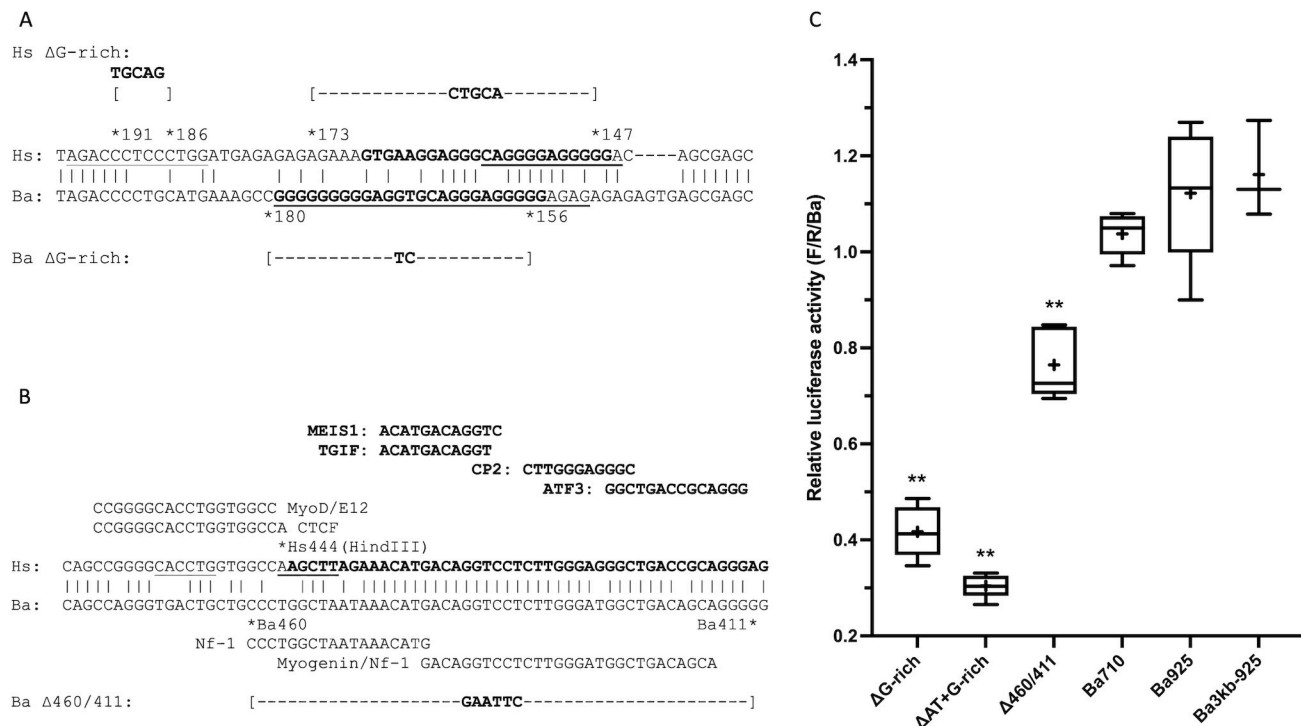

**Fig 6. Sequences and activities of novel regulatory sequences in the Ba gene.** (A) Alignment of the region around the Hs and Ba G-rich sequences. The G-rich sequences Hs168/146 and Ba179/155 are in bold font. The sequences predicted (rVISTA) to bind SP1 are underlined; none of these predicted binding sites is conserved between Hs and Ba. An additional predicted SP1 binding site (MATCH) 5' of the Hs G-rich sequence (Hs195/183) is also underlined. The Ba ΔG-rich mutant replaces Ba180/156 with a TC sequence; the Hs ΔG-rich mutant replaces Hs191/186 with a TGCAG sequence and Hs173/147 with a CTGCA. Analysis of both mutant sequences (rVISTA, MATCH) predicts no SP1 binding sites. (B) Alignment of the Ba449/412 conservation between Hs and Ba. In the Hs sequence, the 5' end of a DNaseI hypersensitive region is in bold text and a nonconserved flanking CTCF binding site is indicated (from UCSC Genome Browser [20], see Fig 7B, Track 4). Above, conserved (rVISTA) sites for transcription factors expressed in muscle (S7 File) are indicated in bold. An E-box (CACCTG) is underlined and nonconserved MYOD and E12 (rVISTA) sites are indicated. A HindIII site (AAGCTT, at position -373 of Devlin et al. [13]) is shown for reference. In the Ba sequence, nonconserved NF1 and composite MYOG/NF1 sites (LASAGNA) are indicated. The Ba Δ460/411 deletion replaces Ba460/411 with a GAATTC sequence. (C) Activity of mutations in the novel Ba regulatory regions. For analysis of statistical significance in this figure, Ba710 was used for comparison; samples indicated by ** had p <0.01 (see S6C File for details). ΔG-rich: Ba ΔG-rich, mean = 42% of control, p <0.0001. ΔAT+G-rich: Ba ΔAT + ΔG-rich, mean = 30% of control, p <0.0001. Δ460/411: Ba Δ460/411, mean = 76% of control, p < 0.0001. 710: Ba710, mean = 104% of control. Ba925: Ba925, mean = 112% of control. Ba3kb-925: mean = 116% of control.

synergistically with the AT element. We therefore combined the AT element deletion, Ba ΔAT, with the G-rich sequence deletion, Ba ΔG-rich, to give Ba ΔAT+ΔG-rich (Ba Δ248/222+Ba Δ180/156, S1B File). The combined deletions reduced expression even further, to 30% of control (Fig 6C). The difference between Ba ΔG-rich and Ba ΔAT+ΔG-rich is statistically significant by two-tail t-test (Ba ΔG-rich (*M* = 0.417, *SD* = 0.054), Ba ΔAT+G-rich (*M* = 0.304, *SD* = 0.025); *t*(8) = 4.255, *p* = 0.003). Since Ba ΔAT alone reduced expression to 68%, and Ba ΔG-rich reduced expression to 42%, it appears that the AT element and G-rich sequence act additively (68% of 42% = 28%), but synergistic interaction is not observed.

The Hs gene also shows predicted (rVISTA) SP1 binding sites in only two places: the CCAC-box and a similarly positioned G-rich sequence (70% G over 23 nt, Hs168/146, Fig 6A). A repeated GAGA motif is present, but in the Hs gene it is 5' of the G-rich sequence. Deletion of the Hs G-rich sequence and an adjacent SP1 site (Hs ΔG-rich, Fig 6A) does not lead to a statistically significant decrease in expression (Fig 5 and S5B File, p = 0.203), although there is a trend toward lower expression (S5A File). Therefore, this region may have weak function, but is clearly not a major determinant of activity in the Hs promoter as it is in the Ba promoter.

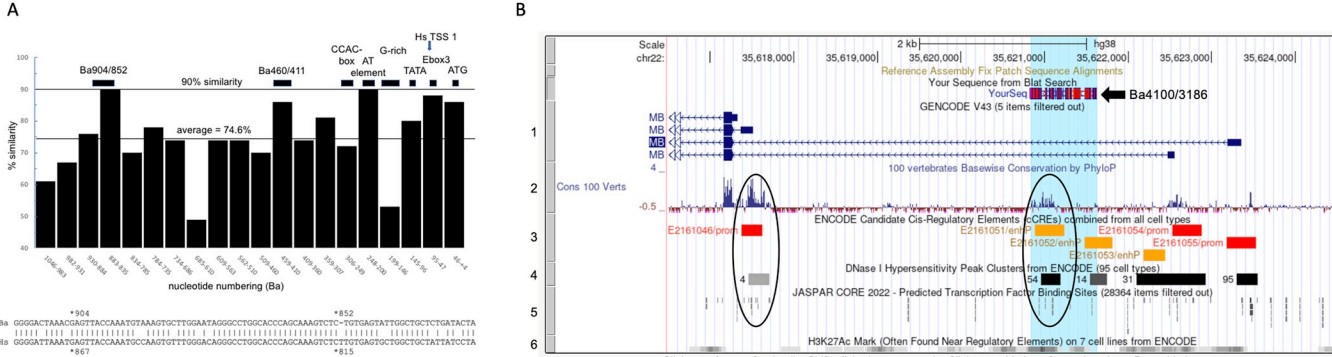

**Fig 7. Two distal regions of high sequence conservation have no discernible activity.** (A) Graph of percent sequence similarity between Hs and Ba from nucleotide +4 (A of ATG = +1) to Ba1046 and Hs996, in blocks of 50 nts relative to the Hs sequence. The average nucleotide similarity across this region is 74.6%. Shown above are regulatory landmarks described in this work. Below is a BLASTn alignment of 100 nt of Ba and Hs sequence around the Ba904/852 conserved region. (B) Presentation of 8,000 nt of human chromosome 22 from the UCSC Genome Browser [20, 21] showing the 5' end of the human myoglobin gene and about 7,000 nt of the 5' flanking region. Selected browser tracks are numbered at left. Above Track 1, arrow, the Ba4100/3186 region tested as the putative "Ba3kb" enhancer is indicated. Circled at the left of the figure is the region shown in Fig 6B: this region has high sequence conservation, a predicted cis-regulatory element (E2161046), a DNase I hypersensitive region (labeled "4"), and a cluster of predicted transcription factor binding sites including a CTCF site. Circled in the center of the figure is the region of high sequence conservation tested as "Ba 3kb", a predicted cis-regulatory element E2161051, a DNase I hypersensitive region, and a cluster of predicted transcription factor binding sites for NFAT, MYF, and MEF transcription factors. Track 1: the 4 most-proximal of the nine known transcription start sites; the one at the top is TSS 1, the major transcription start site, at coordinate chr22:35,617,329; transcription is right to left. Track 2: Sequence conservation across 100 vertebrates. Track 3: Computationally predicted candidate cis-regulatory elements based on ENCODE data. Track 4: DNaseI hypersensitive sites. Track 5: Selected transcription factor binding sites predicted by JASPAR. Track 6: Histone H3 lysine27 acetylation. This UCSC Genome Browser view can be accessed at: https://genome.ucsc.edu/s/csackerson/chr22%3A35%2C616%2C500%2D35%2C624%2C499.

**Conserved sequence at Ba449/412.** Deletion of the Ba promoter to Ba410 reduced expression to 75% of Ba710 (Fig 4C), indicating the region between Ba410 and Ba710 houses additional regulatory elements. In a survey of sequence similarity between Ba and Hs across ~1,000 nt 5' of the MB coding region, an above average region of similarity occurs between Ba460/411 (Fig 7A). Within this region, a stretch of 36/38 conserved nucleotides occurs at Ba449/412 (95% identity, Fig 6B). Since high conservation of non-coding sequence between species can reflect regulatory function [34–37], we explored this sequence further.

In the Hs gene, this region has several notable features (Fig 6B): First, this region is the 5' end of a DNase I hypersensitive region [38] in muscle cells. Second, consistent with this region being a boundary between active (DNaseI hypersensitive) and inactive chromatin, it is flanked by a predicted CTCF [39] binding site (JASPAR, LASAGNA). Third, a cluster of conserved binding sites for transcription factors expressed in muscle is predicted (rVISTA, S7 File). Fourth, this region has an additional E-box identical in sequence to E-box1 (CACCTG) that is predicted (rVISTA) to bind MYOD and E12. In addition, in a comparison between human and mouse sequences, successful alignment between the two species ends near the HindIII site at Hs444 (S6D File). Matches are not found further 5', consistent with this region being the 5' boundary of the MB regulatory region.

The DNaseI hypersensitive status of the Ba gene is unknown, and Ba is not predicted to bind CTCF (LASAGNA), nor does it have the E-box. However, it is predicted to bind nuclear factor 1 (NF1) at two adjacent sites within this region (rVISTA, LASAGNA). NF1 is expressed in muscle (S7 File) and, like CTCF, contributes to boundary functions in chromatin [40]. We made a deletion of this sequence in the context of Ba710 (Ba Δ460/411, Fig 6B), which reduced expression to 76% of control (Δ460/411 in Fig 6C), essentially identical to the reduction in activity seen with the truncated Ba410 promoter (Fig 4C). Deletion of this region in the Hs gene was not conducted.

### Distal conserved sequences do not increase activity

One possible explanation for the low activity of the Ba gene compared to the Hs gene is that activating regulatory elements exist that are not included in the Ba710 construct. Using DNA sequence conservation as a guide, we identified a region at Ba904/852 and Hs867/815 with 91% identity (48/53 nt) between the species (Fig 7A). The only other region with similarly high conservation is at the AT element (59/64 matches, 92% identity). We extended the cloning to a Ba925 endpoint to include this conserved region in our analysis. Although the Ba925 construct increased mean activity to 112% of the Ba710 control (Fig 6C), this is not a statistically significant increase (S6C File).

To gain a genome level view of conservation across the MB locus, we queried the UCSC Genome Browser [20] for information from the ENCODE project [41]. The ENCODE Project has catalogued various markers of candidate regulatory elements in the human genome such as DNA sequence conservation, transcription factor binding, DNase I hypersensitivity, and histone modifications. Our search is shown in Fig 7B. A region of high DNA sequence conservation between multiple species is located ~3.6 kb 5' of the major transcriptional start site in the human sequence. This coincides with a "candidate Cis-Regulatory Element" (cCRE) designated "E2161051/enhP" (Fig 7B, Tracks 2 and 3). This region is DNase I sensitive in the five muscle cell lines tested (Fig 7B, Track 4), and shows moderate histone H3K27 acetylation modifications (Fig 7B, Track 6). cCRE E2161051 also contains predicted binding sites for MYF6, MEF2 and NFAT transcription factors (Fig 7B, Track 5).

cCRE E2161051 maps by alignment to Ba3699/3347. Examination of the surrounding region showed that sequence conservation between Ba and Hs remains high (~80%) for approximately 300 nt further 5'. We cloned a ~900 nt fragment that includes the putative cCRE and the additional conserved region (Ba4100/3186, "Ba 3kb"), and inserted it 5' of the Ba925 vector to create Ba3kb-925. In three paired transfections (Fig 6C, S6B File), Ba3kb-925 and Ba925 were essentially identical in activity by two-tailed t-test (Ba925 ($M = 1.122$, $SD = 0.141$), Ba93kb-925 ($M = 1.161$, $SD = 0.101$); $t(6) = 1.409$, $p = 0.696$). Therefore, despite strong bioinformatics clues of activity of this region, no activity was observed in our assay system.

## Discussion

### Transcription driven by minke whale 5' flanking regions is only 8% that of humans

Cetaceans are notable for the high levels of the oxygen carrying protein myoglobin in their skeletal muscle, and hence myoglobin concentration correlates strongly with diving capability [25, 42]. Traditionally, transcriptional levels have been considered a dominant, although not sole, determinant of protein levels [43–45]. We therefore hypothesized that cetacean myoglobin genes would be highly active transcriptionally, and that comparison of their transcriptional mechanisms with that of humans [reviewed in 17] may provide insights into myoglobin transcriptional control and regulatory evolution.

However, we found that the MB gene from minke whales is only 8% as active as the human gene in a standard in vitro model for muscle cells, mouse C2C12 cells. We expanded our survey to include a total of seven cetaceans and three related, terrestrial artiodactylan species (S2 File), cows, elk, and pigs. We included elk to have a non-domesticated artiodactylan species, since myoglobin levels in domesticated species may be driven by breeding in response to market forces [46]. The ten cetacean and artiodactylan species examined showed, when averaged, an expression level only 6% that of humans (Fig 2, S2 File). Moving further out

phylogenetically, we tested horses and found that the horse gene is >5-fold more active than the human gene, and >90-fold more active than the average of the cetaceans and artiodactylans. Last, we tested a carnivore, dogs, and obtained an activity ~2-fold higher than that of the cetacean and artiodactylan mean (Table 1, Fig 2, S2 File).

This survey demonstrated that transcriptional levels from myoglobin genes can vary greatly, and do not correlate well with myoglobin protein levels. For example, myoglobin protein in human type 1 skeletal muscle fibers is ~4.5 mg/g [47], whereas minke whales contain ~20 mg/g [12]. Yet, despite the 4-5-fold higher protein level in the whale tissue, the minke whale gene is only 8% as active in our assays. There are several possible explanations for this observation.

One possibility is that our cloning missed critical positive cis-acting elements. However, expanding the size of the regulatory region tested to Ba925 to include a conserved region between Ba904/852 did not increase activity to a statistically significant degree (Fig 6C). Further, the addition of 900 nt to the Ba925 construct (Ba3kb-925) to include the candidate cis regulatory element "E2161051/enhP" also did not increase expression levels (Figs 6C and 7). In the work of Devlin et al. [13] on the human gene, ~2000 nt of 5' sequence was tested for activity. Reducing the size of this tested region did not cause a decrease in activity until they encountered what they termed the myoglobin upstream regulatory element (MbURE), beginning at Hs331 (our numbering; -261 in [13]). This same 2000 nt promoter region directs cell type and developmental expression similar to the endogenous MB gene in transgenic mice [48]. Therefore, although further regulatory elements may exist [37], to date, experiments to test for functional regions have not revealed them.

A second possibility is that the trans-acting transcription factors in our in vitro assay system, mouse C2C12 cells, differ in critical ways from those in whales. So, although it does express high levels of transcription from the human and horse genes, it may be unable to express high levels of transcription from the whale genes. Although C2C12 cells have been widely used as a general model for muscle differentiation [49–53], differences exist between C2C12 cells and other popular model systems such as rat L6 cells and cultured human skeletal muscle cells [54]. In addition, C2C12 differentiation only rarely recapitulates the extent of myofiber formation observed in vivo [55]. The transcription factor complement of muscle cells undergoes a well characterized progression between the stem cell, myocyte, and myofiber stages in vivo [55], but it is possible that that progression is less well-ordered in C2C12 cells.

Similarly, physiological signaling, for example from muscle contraction [22, 23], hypoxia [23, 56], nitric oxide [57], or a lipid-rich diet [58, 59], influences transcription factor expression and activity. Our experiments address what we have termed constitutive expression in differentiated, cultured C2C12 cells without the induction of calcium-induced pathways, hypoxia, or other physiological pathways. Therefore, numerous gene regulatory mechanisms may come into play in mature cetacean muscle cells that are not recapitulated in our assay system. Detailed studies characterizing the transcription factors bound to the minke whale gene will be required to resolve this caveat on our results.

In contrast to a transcriptional explanation for the high MB levels in Cetacea, recent studies of MB protein structure and evolution support the idea that the high levels may be due simply to the steady accumulation of a very stable protein. Observations show that neonates have low MB levels compared to adults, and the level of MB increases steadily during the early life of the animal [12, 60, 61]. Several evolutionary adaptations of the MB protein work together to increase MB stability [62, 63] and solubility during synthesis and as mature folded myoglobin, despite the "macromolecular crowding" and acidic pH of the muscle cell cytoplasm [24, 64, 65]. For example, increased positive charge on the surface of the protein [64, 66] has evolved to inhibit aggregation at high protein concentration through electrostatic repulsion. These evolutionary changes permit high cytoplasmic MB levels to accumulate without aggregation

and subsequent destruction [24, 62, 64, 66, 67]. Therefore, high transcription levels are not necessary to account for high MB protein levels in adult cetacean animals.

We have observed a relatively low level of transcription among the Artiodactyla as well as the Cetacea, and even in the carnivore, dogs. This implies that a relatively low transcription level may be the ancestral condition. Evolving an increased rate of transcription to satisfy the physiological need for high MB levels in the absence of protein evolution would not have solved the aggregation problem. Evolving instead a stabilized protein resistant to aggregation is a strategy widely shared among diving mammals [24].

In this context, the high transcriptional activity of the horse gene seems paradoxical. Horse skeletal muscles have MB levels of ~7 mg/g [68], higher than that of humans. Given their access to abundant atmospheric oxygen one might expect the need for intracellular storage in MB to be low. Horse MB has a net surface charge of +1.75 (sperm whale is +4.15 [64]) and therefore aggregates at lower concentrations than sperm whale myoglobin [69], and is 4-fold less stable than minke whale MB [70]. It may be that a high rate of transcription balances a high rate of turnover. Clearly, the horse MB gene has followed a very different evolutionary trajectory than the cetacean genes.

## The minke whale gene has substituted a G-rich sequence for the CCAC-box

We have dissected the 5' flanking regulatory region of the minke whale MB gene, guided by previous studies in humans and mice, and clues from bioinformatics. We find differences in gene regulatory elements between the minke gene and the human gene (Fig 1) consistent with the observed differences in transcription rates. The most striking is the relative inactivity of the minke whale CCAC-box ortholog. Instead, we propose that a G-rich sequence serves a similar function, cooperating with the AT element to activate transcription.

Considerable evidence supports the importance of the CCAC-box in the human gene. In the work of Devlin et al. [13] deletion of the MbURE (to Hs275, -205 in [13]), which contains the CCAC-box, reduced expression to 12%. Additionally, an internal deletion of the region between Hs444 (HindIII, -373 of [13]) and Hs275 in the context of ~1,000 nt of 5' sequence abolished expression. Bassel-Duby et al. [14] created a subtle mutation in the CCAC-box from ACCC to GGTA, referred to as CCAC mut 3 (Hs285/282) that reduced expression to less than 10% of the wild-type sequence. Further, Bassel-Duby et al. [15] and Grayson et al. [18] demonstrate synergistic interaction of the CCAC-box and the AT element.

In contrast, in the minke whale gene, we find that deleting the CCAC-box with Ba ΔSP1-CCAC (Ba334/251) has no effect on expression. This deletion corresponds to Hs323/249, very similar to the 5' deletion of Devlin et al. [13] from Hs331to Hs275 that defined the MbURE. We also recreated the CCAC mut 3 mutation in the minke whale gene, with no effect. Last, combining deletion of the CCAC-box with deletion of the AT element (Ba ΔCCAC-AT) to test for a possible synergy between the two sequences reduces expression slightly compared to Ba ΔAT alone, but this reduction is not statistically significant (p = 0.128, S4C File).

However, changing the minke whale CCAC-box to match the human sequence (CCACswap) increased the expression of the minke whale gene (Fig 4B). Changing the AT element to match the human sequence (ATswap) had no effect by itself (Fig 3B), but combining CCACswap with ATswap increased expression synergistically (CCAC+ATswap, Fig 4B). Thus, even in the background of the minke whale gene, the human CCAC-box has an activating effect in striking contrast to the absence of activity displayed by the minke CCAC-box.

The human CCAC-box is reported to bind the ubiquitous activating transcription factor, SP1 [16]. SP1 is robustly predicted to bind the C-rich CCAC-box, which is 5' of the AT element. SP1 is also predicted to bind at several overlapping sites in a G-rich (C-rich on the other

strand) sequence 3' of the AT element, and in no other regions, in both the human and minke whale genes. When we tested the minke whale G-rich sequence for function, we found that its deletion strongly reduced expression (Fig 6C). In contrast, the human G-rich sequence is non-essential (Fig 5).

Combining the deletion of the minke whale G-rich sequence with deletion of the minke whale AT element (Ba ΔAT+G-rich, Fig 6C) further reduced expression compared to Ba ΔAT alone, and the effect is additive. We therefore propose that the G-rich sequence in the minke whale gene may play the role of recruiting SP1 that the CCAC-box plays in the human gene, but the synergy seen in the human gene between the CCAC-box and the AT element has been lost.

It is well established that cell-type specific gene expression is controlled by combinations of transcription factors bound at promoters and enhancers [71]. Both the human and minke genes are predicted to bind SP1 at their GC-rich sequences (Fig 6, S4A File), as well as MYOD, MYOG, E12 and MEF2 factors at the AT element (Fig 3A), Thus, one might expect similar activity from the two enhancers if they function simply through the accumulation of critical transcription factors. However, the order, distance, and orientation of the transcription factors bound to an enhancer, what has been termed "enhancer grammar", can matter in some cases [72]. Thus, despite a similar constellation of factors predicted to be bound in the minke whale and human genes, changing the spatial relationships between these transcription factors may contribute to the lower transcriptional output. The loss of synergy described above would be an example of this effect.

Both the CCAC-box and the G-rich sequence are poorly conserved between humans and minke whales (Figs 4A, 6A and 7A). Repetitive sequences are prone to rapid change, through DNA slippage during replication [73–76]. Sequence flexibility in these runs of G-C nucleotides may have facilitated the proposed evolutionary switch between the CCAC-box and the G-rich sequence.

We have presumed that the CCAC-box and the G-rich sequence function primarily by binding SP1. Considerable experimental evidence supports this bioinformatics assignment [16, 77, 78]. However, alternative explanations exist for the activity of the human CCAC-box and minke G-rich sequence. Runs of G have been shown to form G-quadruplex structures that are highly enriched at promoters [79] where they may mediate a variety of functions including epigenetic regulation. G-quadruplex structures have also been shown to bind MYOD [80]. Indeed, MYOD binding to a non-E-box site in the human CCAC-box is predicted (S4A File). In contrast, MYOD binding to the minke whale CCAC-box is not predicted. In addition, when the CCAC mut 3 mutation of Bassel-Duby et al. [14] is examined (rVISTA) MYOD binding is not predicted, but SP1 binding is still predicted (Fig 4 legend). Last, the human CCAC-box has been shown to bind a nuclear factor of 40 kD [14], the approximate size of MYOD (see, for example, [81]; 34.5 kD in Uniprot [82]). This 40 kD protein is described as being distinct from SP1 [14]; consistent with this, SP1 has a molecular weight of approximately 80.7 kD [82, 83]. Thus, the human CCAC-box may have functions beyond the binding of SP1 that contribute to its relative importance, compared to the minke CCAC-box. Similarly, the minke G-rich sequence may function through mechanisms beyond SP1 binding.

## Functioning of the MEF2 and E-box sites has evolved between the human and minke whale genes

At the core of the AT element is a MEF2 binding site [14, 15, 84] flanked on both sides by E-boxes, E-box1 and 2 (Fig 3A). An additional E-box, E-box3, occurs immediately 3' of the transcriptional start site (Fig 3C). The functions of these sites differ between humans and minke whales.

In previous studies of the human AT element [14, 15] mutation of the MEF2 site reduced expression to 20% of wild-type. We used the same nucleotide changes in the minke whale gene, but expression was only reduced to 88% of control, a reduction that was not statistically significant. This mutation is predicted to no longer bind MEF2, and this is the only predicted MEF2 binding site in the Ba710 sequence.

This difference is unlikely to be due to a lack of MEF2 activity in our assay system. MEF2 has been shown to be expressed in C2C12 cells by three days of differentiation [85]. Although the previous studies on the human MB gene used the mouse skeletal muscle cell line Sol8 [14] or mouse and rat heart muscle [15], electrophoretic mobility shift assays comparing DNA binding activities of nuclear proteins from sol8 and C2C12 myotubes appear identical [18]. Also, although MEF2 is activated by increases in intracellular calcium via the calcineurin phosphatase [86], MEF2 can activate gene expression in C2C12 cells after differentiation in the absence of specific manipulation of calcium flux [87, 88]. Therefore, MEF2 is likely to be both present and capable of activating gene expression in our assay system.

Mutations of the minke whale E-boxes also had different results from those reported for the human E-boxes. In the human gene, E-box1 was previously shown to be nonessential but E-box2 was required for full activity [14]. In the minke whale gene, mutation of either E-box1 or E-box2 had no effect in our experiments. Only when the entire AT element is deleted, removing both E-boxes and the MEF2 site, is there an effect on expression (Ba ΔAT, Fig 3B). It is possible that the components of the minke whale AT element serve redundant functions such that loss of any one component has no effect.

It is possible that E-box3 plays a role in these differences. Mutation of E-box3 showed a significant decrease in expression (Ba E-box3 mut, Fig 3B), whereas E-box3 mutation in the human promoter is reported [27] to increase expression. E-box3 is of the "symmetrical" type (CAGCTG) [89] capable of binding MYOD as a homodimer, as well as MYOD-E12 heterodimers [90]. Binding of MYOD is robustly predicted for E-box3 for both humans and minke whales (Fig 3). To explain the results with the AT element, one could imagine that interactions between MYOD bound to E-box3 and MEF2 [91–93] could mask the loss of E-box1 or 2, or E-box3 could interact with E-box1 or 2 to activate expression in the absence of MEF2 [94]. Intriguingly, MYOD-E12 heterodimers bound to an E-box can interact with SP1 and serum response factor (SRF) to activate gene expression [95, reviewed in 96]. As described, SP1 binding is broadly predicted in the minke promoter-proximal sequences. Binding of SRF is predicted (LASAGNA, MATCH, rVISTA) in only two places in the Ba710 minke whale sequence: within E-box2 (CCATGTGAGG, Ba228-219) and immediately upstream of E-box3 (CCCTTTAGGGCCA, Ba94-82). SRF is not predicted (rVISTA) to bind the Hs671 human gene. Thus, the mechanisms of activation of the minke whale gene may differ in numerous ways from the human gene.

## Novel regulatory sequences are found in the minke gene

Our explorations of minke whale MB regulation revealed two functional regions that had not been noted in studies of the human gene. The first is the G-rich sequence discussed above. The second is a conserved sequence at Ba449/412 (tested as a larger deletion of Ba460/411). In the human gene, the features of this region include the 5' end of a DNase I hypersensitive region and a CTCF binding site (Fig 6B). These observations are consistent with this region being the 5' boundary of a functional regulatory element [97, 98]. With regard to the minke whale gene, CTCF is not predicted to bind, but two NF1 binding sites are predicted (Fig 6). NF1 has been shown to have insulator activity, delineating a boundary between chromatin domains [40], much like CTCF. Thus, although the minke region is not predicted to bind CTCF, and the

human region is not predicted to bind NF1, the functional consequences may be the same. The Ba460/411 deletion removes not only the NF1 sites, but also binding sites for muscle transcription factors conserved between humans and minke whales, and this caused a significant loss of activity. It is unknown why the activity of this region was not uncovered in previous studies of the human gene [13]. Further exploration of the role of this region in MB gene control may be productive.

### Limitations of the study

The approaches taken in these studies have several inherent limitations. First, the assays for function were conducted by transient transfection into a cultured cell line growing in vitro. One limitation of this approach is that the transfected gene lacks its native genomic context such as neighboring genes, transcriptionally associated domains, and regulation by chromatin [99]. However, transfection has been a mainstay of gene expression studies for nearly 50 years [100]. As we have demonstrated, transfection can provide valuable first steps in understanding previously unknown aspects of gene regulation.

Cultured cells growing in vitro also lack context, in this case the complex physiology of intact muscle tissue. For example, two previously identified NFAT transcription factor binding sites [17, 101, 102] are found in the human promoter. The NFAT transcription factor is activated by the increased calcium accompanying muscle contraction, coupling muscle activity to the expression of proteins involved in contractility [102, 103], and activating myoglobin expression [32]. Three putative NFAT binding sites can be identified in the minke 5' region (at Ba319, Ba588, and Ba810), but their deletion does not cause changes in expression. Calcium concentration is not manipulated in our assays, so the effect of NFAT is not addressed. Similarly, other physiological influences would not be detected.

A second limitation is that our experiments were done in mouse cells, and the trans-acting transcriptional machinery in mouse cells may not be an appropriate model for activities in whale cells, as discussed above. For example, the activity, or lack thereof, of the CCAC-box in whales, in vivo, must technically be considered unknown, despite our results. Similarly, definition of the human CCAC-box was based on experiments in chick [13], mouse [14], and rat [15] cells, so the same may be said of the human CCAC-box. It will be of interest to clarify this point once cetacean-derived muscle cell lines are available [104].

Last, our studies have relied on bioinformatics-based signals of function, such as transcription factor binding predictions and inter-species DNA sequence conservation. Despite the promise of such approaches, the majority of "hits" must necessarily be false positives [reviewed in 105]. For example, we found the conserved region at Ba460/411to have function, but the conserved regions at Ba904/852 and at the -3kb region did not affect transcription in our assays (Figs 6C and 7).

### Conclusions

Our most striking observation in these studies is the low constitutive expression of the minke whale myoglobin gene, in contrast to the high myoglobin protein levels seen in adult muscle tissue [4, 8]. We propose that the low transcription level is the result of multiple evolved regulatory differences between the human and minke genes. Low transcription levels may be acceptable because of the evolution of increased stability of the myoglobin protein [24, 62–65, 106], allowing steady accumulation of the protein consistent with observations in vivo [12, 60]. These findings also imply that the induction of transcription by physiological influences such as exercise [32, 103] and diet [107] may be important in the ontogeny of the high muscle myoglobin protein levels seen in cetaceans.

## Materials and methods

### Species survey tissue and genomic DNA samples

Total genomic DNA was isolated from tissue samples using DNeasy Blood & Tissue Kit (Qiagen, catalog # 69504).

Cetacean tissue samples were collected from stranded, deceased animals by member organizations within the National Marine Mammal Stranding Network, The Hawaii Pacific University Stranding Program (HPUSP), the Oregon Marine Mammal Stranding Network (OMMSN), the International Fund for Animal Welfare (IFAW), and the Alaska Stranding Network (ASN) under the authority of a National Marine Fisheries Service (NMFS) Stranding Agreement issued to each of the cooperating organizations by the Office of Protected Resources, National Oceanic and Atmospheric Administration. Laboratory use of the tissues was conducted under a Marine Mammal Parts Handling Authorization, provided by regional offices of NMFS. As this prior review had been conducted by experts in this field further ethical review by the cooperating institution, California State University Channel Islands, was not required.

Human DNA was isolated from a saliva sample provided by CS. Cow and pig DNA was isolated from locally purchase beef and pork. Elk DNA was isolated from elk steak purchased from Basspro.com (catalog # 1568645).

Horse (catalog # GE-170) and dog (catalog # GD-150M) DNA was purchased from Zyagen. com.

### Myoglobin gene cloning and preparation of plasmids used for transfection

See S8 File for details of clonings and primers used for each reported construct. In general, MB gene sequences were isolated through two rounds of PCR using Accuprime Pfx DNA polymerase (Invitrogen, cat # 12344–024). The first round used as the 5' (forward) primer "bosF1", a degenerate primer designed from an alignment of human, horse, and cow sequences (beginning at a nucleotide equivalent to Ba1024 for reference). The 3' (reverse) primer "stenR3" was designed from the dolphin *Stenella attenuata* exon 1 sequences (beginning at the equivalent of nt Ba+65). The PCR products of this first round were typically complex, so a second round of PCR was carried out using nested primers. In the second round, various 5' primers were used (S8 File). Two 3' primers, "stenR1" and "stenR2", were used depending on which gave a unique product; these primers were designed from *S. attenuata* exon 1 sequences (beginning at Ba+3 or Ba+51, respectively). The blunt-ended round two product was sufficiently pure for cloning into pIBI31 (addgene.org/vector-database/3128) cut with SmaI and selected as white colonies on Xgal plates. The clone was moved into pGL4.10[luc2] (Promega, catalog # E6651) with XhoI and NcoI. The result of this strategy is the inclusion of 30 nt of pIBI31 polylinker sequence at the 5' end of the pGL4.10 clone. All the species tested have their ATG in a CCATGG sequence recognized by NcoI, so the 3' end was a direct fusion of the cloned ATG translational initiation sequence to the firefly luciferase ATG of pGL4.10[luc2].

As genomic sequences became publicly available, subsequent clonings and manipulations used species-specific 5' (forward) primers with an XhoI site the 5' end to allow direct cloning into pGL4.10[luc2] without extraneous polylinker sequences, as detailed in S8 File. This difference was not seen to affect expression in a systematic way.

The pSV-Rluc renilla luciferase internal standard vector was derived from psiCHECK-2 (Promega, catalog # C8021) by digestion with EagI to remove the firefly luciferase gene, leaving the renilla luciferase gene driven by the SV40 promoter.

All pGL4.10[luc2] derivatives used for transfections were verified by sequencing (S8 File).

For transfection, plasmids were purified on QIAprep Spin Miniprep columns (Qiagen, catalog # 27104) with an extra wash with PB binding buffer, and quantitated on a NanoDrop ND-1000 spectrophotometer.

## Cell culture, transfection, and luciferase assays

Mouse C2C12 cells [108] used were obtained from ATCC (catalog # CRL-1772), except three experiments for which the cells used were obtained from the laboratory of Barbara Wold at CalTech (originally from ATCC), as indicated in S2 File.

For routine passage, cells were plated at $5 \times 10^4$ cells in 60mm tissue culture plates in "growth medium": DMEM (high glucose, without pyruvate; Thermo-Fisher cat # 11965092) with 10% fetal bovine serum (Gibco cat # A3160501); no antibiotics were used. Cells were passaged every two days. All experiments were performed with cultures that had gone through fewer than 10 passages since receipt.

For transfection, cells were plated in 12-well tissue culture plates at $1 \times 10^5$ cells per well in 1.2 ml growth medium. Immediately after plating, the transfection mix was prepared and added to the cells, within about 1 hour of plating. Transfections used 1 µg pGL4.10-derived reporter plasmid plus 1 ng pSV-Rluc renilla luciferase for normalization (see below) in a total volume of 50 µl with DMEM (no serum). 5 µl Polyfect Transfection Reagent (Qiagen, catalog # 301105) was added, vortexed for 10 seconds, incubated 10 minutes, and diluted with 300 µl growth medium; the entire volume was added to each well. In practice, duplicate wells were transfected from a master mix scaled up 2.2-fold from the above quantities and 355 µl used for transfection. After 24 hours, the media was replaced with DMEM with 2% horse serum (Fisher, catalog # 301105) and ITS (Gibco, catalog # 4140045) at 1:1000 dilution ("differentiation medium"); this is day zero of differentiation. Media was refreshed every day until the cells were harvested.

In initial experiments included in the species surveys, each sample was transfected in triplicate, and differentiation was carried out for 6 days. These parameters were changed to 4 days of differentiation to avoid cell detachment, and samples were transfected in duplicate for all later experiments. Differentiation was extensive after 4 days of differentiation judged by morphology and expression of the slow isoform of the myosin heavy chain (S9 File). The change from 6 days to 4 days was not seen to affect the results after normalization to a Ba710 control included in duplicate in each transfection plate (that is, each 12-well culture plate included 2 wells of the Ba710 control plus 5 experimental samples in duplicate). In all cases replications were averaged, and this average represents an n = 1.

Assays used the Dual-Luciferase Reporter Assay System (Promega, catalog # E1910). Each transfected well was lysed into 200 µl Passive Lysis Buffer, and 50µl was used to measure firefly activity with 50 µl LARII, followed directly by 50 µl Stop&Glo to measure renilla luciferase activity, on a FilterMaxF5 microplate reader (Molecular Devices) using a 96-well opaque white microplate. For the horse (Ec) samples, 20µl of lysate was used to ensure the sample did not exceed the range of the plate reader.

## Data analysis and statistics

Luciferase activity was processed as follows: firefly luciferase counts for each well were normalized to the co-transfected renilla luciferase control to give "F/R". F/R was also calculated for the two Ba710 samples included in each plate as a control and averaged to give "Ba". Then, for each well, F/R was divided by the average of the Ba710 control to give "F/R/Ba"; the duplicates in each plate were averaged and this number is the result from each transfected plate. Descriptions of activity as "x% of control" refer to the F/R/Ba value. See S10 File for the data from each

transfection used and an example of the normalization procedure. For Fig 6D, independent experimental sets of Ba710 transfections were processed in the same way to generate an independent experimental Ba710 sample.

Statistical analysis and graphing was done with GraphPad Prism (version 9), using standard parametric tests. For multiple comparisons, ANOVA was followed by Tukey's HSD test [109]. Data presented in each figure was analyzed as a set, using one of the samples with a mean ~1.0 for post-hoc Tukey comparisons, as specified in the figure legend. In cases where pairwise comparisons were made, a two-tailed t-test was used; $p < 0.05$ was considered significant, with further consideration for multiple identical tests ($\alpha = k/n$).

## Bioinformatics

We searched for transcription factor binding sites in Ba sequences using online search programs: MATCH [110] using the "muscle_specific.prf" profile, LASAGNA [111], and rVISTA [112]. rVISTA queries two sequences to look for conserved binding sites. Transcription factor predictions in the human sequence presented in the UCSC Genome Browser interface used JASPAR [113]. Assignment of transcription factor expression in muscle cells, described in figure legends and S7 File, is according to https://www.genecards.org [114] and www.proteinatlas.org [115]. Dot-plot alignment (S6D File) is from PipMaker [116].

## Supporting information

**S1 File. Supplement to Fig 1. A** Flow chart of the experiments. Created under license from BioRender.com. See S9 File for photographs of differentiated C2C12 cells after 4 days. **B** Schematics of the internal deletions described in the text. To the right is the name of the deletion and in parentheses the nucleotides deleted or otherwise affected in each case. Except for the two Ba410 derivatives, deletions are in the context of Ba710.
(DOCX)

**S2 File. Full data set for Table 1.** Data is expressed as F/R/Ba (see Materials and Methods). Four species not shown for simplicity in Table 1 or Fig 2 are included here: *Balaenoptera musculus*(Bm706), *Megaptera novaeangliae* (Mn704), *Orca orcinus* (Oo703), and *Sus scrofa* (Ss657). Data indicated by a superscript "x" is from clones inserted directly into the XhoI and NcoI sites of pGL4.10 without extraneous polylinker sequences. Data generated in C2C12 cells from the Wold lab are indicated by a superscript "w"; all other data was generated in C2C12 cells purchased directly from ATCC. The average of the seven cetacean species shown is 0.801; including the three Artiodactylan species yields an average of 0.782 and this number is used for Fig 2B (CetArt). "SEM" is standard error of the mean.
(DOCX)

**S3 File. Supporting information for Fig 3. A** Alignment of the AT element (E-box1—Mef-2 —E-box2): Hash marks indicate identity with the human (Hs) sequence. Most cetaceans (Mn, Er, Bm, Dc, Oo) and the artiodactylans (Bt, Cc) are identical to Ba through the AT element; Pp has a variant E-box1 (CACATG instead of CACCTG). **B** Alignment of E-box3 region: The Odontoceti, Oo (*Orca orcinus*), Dc (*Delphinus capensis*), Tt (*Tursiops truncatus*), and Pp (*Phocoena phocoena*) vary from the CAGCTG present in the other species presented. The bottlenose dolphin, *T. truncatus*, is included because no published sequence is available for the common dolphin, *D. capensis*. Mm: *Mus musculus*. The variant sequence in the Odontoceti is not predicted by rVISTA to bind any of the transcription factors shown in Fig 3C. **C** Full data set, average of duplicate wells, normalized as F/R/Ba. Equal variances confirmed, based on homogeneity of variances test (indicated here and below with an asterisk on the construct

name). Analysis of variance (ANOVA) confirms a statistical difference between the samples ($F$ (5,24) = 24.955, $p < 0.001$). ANOVA was followed by the post-hoc Tukey HSD test [35]. **D** Tukey test data for Fig 3B.
(DOCX)

**S4 File. Supporting information for Fig 4. A** Multiple species alignment of the CCAC-box region: The 10 nt conserved core of the CCAC-box [15] encompasses Hs289/280 and Ba297/288. Hash marks indicate identity with the human (Hs) sequence. Sp1 binding (bold type) to the Ba sequence is robustly predicted by rVISTA, LASAGNA, and MATCH, and to the Hs sequence by rVISTA, MATCH and JASPAR; Sp1 binding is also predicted by rVISTA for Pp, Ss, and Cf, but not for Bt or Ec. Other transcription factors that are predicted by rVISTA (but not necessarily conserved) and are expressed in muscle are shown: AP2 (asterisks), MAZR (colon), MYOD (wavy underline), PPARalpha (double dashed underline), and USF (underline). **B** F Full data set, average of duplicate wells, normalized as F/R/Ba. Equal variances confirmed, based on homogeneity of variances test. ANOVA confirms a statistical difference between the samples ($F(7,35) = 99.449$, $p < 0.001$). ANOVA was followed by the post-hoc Tukey HSD test. **C** Tukey test for Fig 4B and accompanying text. **D** Full data set for Ba410 and Ba410ΔCCAC, average of duplicate wells, normalized as F/R/Ba. Variances between these two samples are not equal.
(DOCX)

**S5 File. Supporting information for Fig 5. A** Full data set, average of duplicate wells, normalized as F/R/Ba. Equal variances confirmed, based on homogeneity of variances test. ANOVA confirms a statistical difference between the samples ($F(3,15) = 16.109$, $p < 0.001$). ANOVA was followed by the post-hoc Tukey HSD test. The data in heavy boxes is derived from four transfected plates, allowing direct comparison of the activities of Hs671, Hs ΔAT, Hs ΔCCAC, and Hs ΔG-rich; note that although the data for Hs ΔG-rich is not statistically different from that for Hs671, in each case the Hs ΔG-rich samples are lower than the Hs671 values. The same Hs671 data shown here was included in the data set in Table 1. **B** Tukey test for Fig 5.
(DOCX)

**S6 File. Supporting information for Fig 6. A** Full data set, average of duplicate wells, normalized as F/R/Ba. Equal variances confirmed, based on homogeneity of variances test. ANOVA confirms a statistical difference between the samples ($F(3,15) = 151.582$, $p < 0.001$). ANOVA was followed by the post-hoc Tukey HSD test. **B** Full data set, average of duplicate wells, normalized as F/R/Ba. The data in heavy boxes is derived from three transfected plates, allowing direct comparison of the activities of Ba710, Ba925, and Ba3kb-925. Equal variances confirmed, based on homogeneity of variances test. Analysis by ANOVA fails to find a statistical difference between the samples ($F(3,15) = 1.152$, $p = 0.360$). ANOVA was followed by the post-hoc Tukey HSD test. **C** Tukey test for Fig 6C. **D** Left: Dot plot alignment (PipMaker [98]) of 1000 nt of mouse (*Mus musculus*, Mm) MB sequence 5' of the ATG (Y-axis) against 1000 nt of human MB sequence 5 of the ATG (X-axis). The diagonal dashed line plots similarities between the two sequences using default settings. The 5' end of the similarities occurs at nts Hs474 and Mm422. The HindIII site at Hs444 is shown for reference. Right: Dot plot alignment of 1000 nt of Ba MB sequence 5' of the ATG (Y-axis) against 1000 nt of Hs MB sequence 5' of the ATG (X-axis) shown for comparison. **E** Multiple species alignment of conserved sequences at Ba460/411. Vertical hash marks indicate identity with the Hs sequence. The HindIII site (AAGCTT) is bold for reference. Transcription factor sites conserved (rVISTA) with humans and expressed in muscle are indicated: ATF3 (underlined), MEIS1-TGIF (double

underline), CP2 (asterisk), NFE2L1/TCF11 (dashed line), TTF1 (dotted).
(DOCX)

**S7 File. Sources for transcription factors' expression in muscle.**
(DOCX)

**S8 File. Supporting information for materials and methods. A** Details of the clonings used for the species scan. **B** Published sequence accession numbers. **C** Primers used for clonings. **D** Primers and polymerase reagents used for mutations and deletions.
(DOCX)

**S9 File. Differentiation of C2C12 cells under transfection conditions.** Differentiation of C2C12 cells over 4 days in differentiation medium. Top row: Phase contrast pictures of trans-fected cells. Bottom row: Untransfected cells stained for myosin heavy chain; primary antibody was mouse monoclonal anti-myosin (skeletal, slow) from Sigma-Aldrich (cat # M8421, RRID: AB_477248) at 1:1000 dilution; detection with horse radish peroxidase used Vector ImmPress kit (cat # MP-7402).
(DOCX)

**S10 File. Transfection data in Excel format. A** Data. Full data set, organized by transfected plate number. Shown is the average of duplicate wells in each transfection, normalized as F/R/Ba (see S10B File). Samples used in the figures are highlighted in bold, red font. **B** Example. Illustration of the data processing. Raw firefly counts (F counts) are normalized to the renilla control (R counts) for each well to give "F/R". The two Ba710 control samples in each plate are averaged to give "Ba". Each F/R value is then normalized to the Ba value to give "F/R/Ba". The duplicate F/R/Ba values are averaged to give the activity of each sample for that transfection. This number is used in further analysis as an "n" of 1.
(XLSX)

## Acknowledgments

We thank the National Marine Mammal Stranding Network, The Hawaii Pacific University Stranding Program (HPUSP), the Oregon Marine Mammal Stranding Network (OMMSN) the International Fund for Animal Welfare (IFAW) and the Alaska Stranding Network (ASN) for supplying the cetacean tissues. We would like to thank members of the CSUCI Biology Department, especially Mike Mahoney, Cathy Hutchinson, and Jessica Dalton for keeping things running. We would also like to thank Chunnian Zhao for introducing us to luciferase assays, Cori Newton for her enthusiasm and encouragement, and Margaret Karow and the reviewers for their thoughtful comments, which greatly improved the manuscript. We are grateful to Brian Williams and the Wold Lab for the generous gift of C2C12 cells. Last, we rec-ognize the many students that have contributed to this project over the years in the context of the Biology Department Independent Research (Biol 494) course.

## Author Contributions

**Conceptualization:** Charles Sackerson, Rachel Cartwright.

**Data curation:** Charles Sackerson.

**Formal analysis:** Charles Sackerson, Rachel Cartwright.

**Funding acquisition:** Charles Sackerson, Rachel Cartwright.

**Investigation:** Charles Sackerson, Vivian Garcia, Nicole Medina, Jessica Maldonado.

**Methodology:** Charles Sackerson.

**Project administration:** Charles Sackerson.

**Resources:** Charles Sackerson, Rachel Cartwright.

**Supervision:** Charles Sackerson.

**Validation:** Charles Sackerson, Vivian Garcia, Nicole Medina, John Daly.

**Visualization:** Charles Sackerson.

**Writing – original draft:** Charles Sackerson.

**Writing – review & editing:** Charles Sackerson, Vivian Garcia, Nicole Medina, Jessica Maldonado, John Daly, Rachel Cartwright.

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
