## [Decision Letter · Decision Letter 0]

9 Jun 2023

PONE-D-23-10581Comparative analysis of myoglobin in Cetaceans and humans reveals novel regulatory elements and evolutionary flexibilityPLOS ONE

Dear Dr. Sackerson, 

Thank you for submitting your manuscript to PLOS ONE. After careful consideration, we feel that it has merit but does not fully meet PLOS ONE’s publication criteria as it currently stands. Therefore, we invite you to submit a revised version of the manuscript that addresses the points raised during the review process.

In particular, we would expect a revised manuscript to address the potential impact of trans-acting elements in the experiments. The discussion should be revised to include more references, to discuss the potential linkage between the promoter activity of the whale myoglobin and the slow myoglobin accumulation in cetaceans, and the pros and cons of improving transcription rate or protein stability. This is not to say, however, that we consider any other concern raised by our referees to be any less important.

We look forward to receiving your revised manuscript.

Kind regards,

Julie Dumonceaux

Academic Editor

PLOS ONE

Journal Requirements:

“We are indebted to the National Marine Mammal Stranding Network, The Hawaii Pacific University Stranding Program (HPUSP), the Oregon Marine Mammal Stranding Network (OMMSN) the International Fund for Animal Welfare (IFAW) and the Alaska Stranding Network (ASN) for supplying the Cetacean tissues. We are also indebted to CSUCI and the Biology Department, and especially its’ Chairs, for generous provision of space and support; and the Biology Support staff, especially Mike Mahoney, Cathy Hutchinson, and Jessica Dalton. Last, we recognize the many students that have contributed to this project over the years in the context of the Biology Department Independent Research (Biol 494) course.”

 “This work was supported by internal grants from California State University Channel Islands to CS: Research, Scholarship and Creative Activity Grant #811-GD970 (2016), Faculty Research and Development Minigrant (2016), Student Research Steering Council Microgrant (2015), Provost’s Faculty Resource Fund Grant #GD945 (2011). Additional support came from instructional funding for Independent Research (Bio 494). The funders had no role in study design, data collection and analysis, decision to publish, or preparation of the manuscript.”

Reviewers' comments:

Reviewer's Responses to Questions

**Comments to the Author**

1. Is the manuscript technically sound, and do the data support the conclusions?

Reviewer #1: Yes

Reviewer #2: Yes

Reviewer #3: Yes

Reviewer #4: Partly

2. Has the statistical analysis been performed appropriately and rigorously? 

Reviewer #1: Yes

Reviewer #2: Yes

Reviewer #3: N/A

Reviewer #4: Yes

3. Have the authors made all data underlying the findings in their manuscript fully available?

Reviewer #1: Yes

Reviewer #2: Yes

Reviewer #3: Yes

Reviewer #4: Yes

4. Is the manuscript presented in an intelligible fashion and written in standard English?

Reviewer #1: Yes

Reviewer #2: Yes

Reviewer #3: Yes

Reviewer #4: Yes

5. Review Comments to the Author

**Reviewer #1**: The data are solid and interpreted responsibly within the limited scope of the model system. The findings are insufficient to reveal a basis for the abundance of myoglobin in cetaceans compared with terrestrial mammals.

**Reviewer #2**: 1. The manuscript described technically sound and the data support the conclusions.

2. The statistical analysis have been performed appropriately and rigorously.

3. The authors made all data underlying the findings in their manuscript fully available.

4. The manuscript presented in an intelligible fashion and written in standard English.

5. The reviewer does not have any further comments other than the attached "Comments to the authors" file.

**Reviewer #3**: The present manuscript provides an impressively comprehensive molecular dissection of the upstream promoter region of a baleen whale myoglobin gene. The finding that the promotor activity of the whale myoglobin gene is <10% that of humans is initially somewhat surprising given the much higher (~5 to 40 fold higher) maximal myoglobin levels found in cetacean locomotor muscle relative to humans. However, this finding compliments previous (and puzzling) research on the ontogeny of myoglobin accumulation in cetaceans. Briefly, it has long been known that—despite its importance for supporting and extending long submergence times—myoglobin levels only develop slowly in cetaceans, though the underlying reason(s) for this protracted increase has been (to my knowledge) unexplored. I thus highly recommend that the authors briefly discuss this potential linkage in the discussion section as it will (in my opinion) increase the value of the study to comparative diving physiologists. To assist with this, I have appended a few recent papers that highlight the protracted development of myoglobin stores in whales as a starting point.

Cartwright, R., Newton, C., West, K.M., Rice, J., Niemeyer, M., Burek, K., Wilson, A., Wall, A.N., Remonida-Bennett, J., Tejeda, A. and Messi, S., 2016. Tracking the development of muscular myoglobin stores in mysticete calves. PLoS One, 11(1), p.e0145893.

Noren, S.R. and West, K., 2020. Extremely elevated myoglobin contents in the pelagic melon-headed whale (Peponocephala electra) after prolonged postnatal maturation. Physiological and Biochemical Zoology, 93(2), pp.153-159.

Noren, S.R., 2023. Building cetacean locomotor muscles throughout ontogeny to support high performance swimming into adulthood. Integrative and Comparative Biology, p.icad011.

To assist with the revision, I also provide additional comments below that should be addressed/corrected.

Minor comments.

General. While it is correct to capitalize the first letter of “Cetacea”, the term “cetaceans” is not a proper noun and should not be capitalized (same goes for “artiodactylan” and common names for other

Line 54. What is meant by “Traditionally, evolution has focused on changes in amino acid sequence”? Do you instead mean ‘research’ has focused on changes in amino acid sequence? Please clarify.

Line 154. “the Cetacea”  “mysticetes” (or baleen whales).

Line 226. “ar edispensable”  “are dispensable”

Lines 522-531. Another important consideration here is the newly synthesized, heme-lacking and unfolded (or partially folded) myoglobin chains – apomyoglobin. Briefly, an increased net surface charge may also be important for reducing aggregation of these nascent protein chains, thereby permitting more efficient heme uptake while also helping to ensure that apomyoglobin folding outcompetes its rate of precipitation (see reference 26 for more information on this) leading to more mature myoglobin proteins being formed.

Lines 707-710. The finding that the myoglobin gene of ungulates and pigs also have very low constitutive expression levels instead argues that this trait is ancestral to the increase in net surface charge (ZMb) seen in cetaceans. In short, there is no evidence to suggest that these regulatory differences co-evolved in concert with the increase in ZMb in cetaceans without a similar analyses of hippo, ungulate, and suid promoter regions. Notably, given the propensity of nascent myoglobin/apomyoglobin proteins to precipitate, increasing the gene activity in the absence of an increase in ZMb might be counter productive. This insight is important, as it also helps explain the long perplexing observation that myoglobin stores develop very slowly in whales, despite the importance of this oxygen-binding protein for extending the underwater endurance of members of this group.

**Reviewer #4**: I pasted comments with Character Count 3258 to this window, but your system insists "Minimum Character Count Not Met" and I could not proceed. Please use the comment in the attached file. This sentence should be ignored.

6. PLOS authors have the option to publish the peer review history of their article (what does this mean?). If published, this will include your full peer review and any attached files.

Reviewer #1: No

Reviewer #2: No

Reviewer #3: No

Reviewer #4: **Yes: **Tsuyoshi Shirai

---

## [Author Response · Author response to Decision Letter 0]

19 Jul 2023

Response to reviewers:

Academic editor:

• Re: Trans-acting elements: This point was raised by the reviewer of “Comment_230603” and has been addressed as described in the response to that reviewer, below.

• Re: More references: Suggested references were provided by two reviewers. The reviewer of the “Comments to the authors” (reviewer #2) suggested papers by Isogai et al and Berenbink, and these have been incorporated as described in the response to that reviewer, below. Also, reviewer #3 suggested papers by Cartwright and Noren; these have been included as described in the response to that reviewer, below. 

• Re: The potential linkage between the promoter activity of the whale myoglobin and the slow myoglobin accumulation in cetaceans: This point is raised by reviewer #3 and has been addressed as described in the response to that reviewer, below.

• Re: The pros and cons of improving transcription rate or protein stability: This issue, too, is raised by reviewer #3. The comments of reviewer #3 have led to considerable revision of the Discussion section titled “Transcription driven by minke whale 5’ flanking regions is only 8% that of humans”. We originally had discussed “post-transcriptional mechanisms”, which would imply regulatory mechanisms operating in real time in response to conditions (e.g., increases in mRNA stability or in translational rate). Instead, reviewer #2 and #3’s comments have steered us toward protein evolution. In our opinion this has greatly improved this portion of the discussion, and better addresses some of the thinking in the field that we had not given sufficient attention.

Re: Journal Requirements:

• The initial review indicated that the style requirements have been followed.

• 2, 3. The wording in the Acknowledgements section was not intended to imply funding. The wording has been changed. The Funding statement is not changed. The new sections are in the cover letter.

• 4. Regarding data availability, all data has been included in the manuscript and supplemental files, as stated in the Data Availability section. S10 File has been added to include the data and an example of how raw data was processed in an accessible Excel format.

• 5. Supporting information files have been listed at the end of the manuscript. Included are the name in the form of “S1 File.docx”, a title, and a legend.

Additional notes:

Two changes to the submitted material have been made that are not in response to the editors or reviewers’ comments:

• S4 File: A Tukey test comparing Ba ∆AT, Ba ∆CCAC, and Ba ∆CCAC-AT has been added to S4C. Previously, a T-test was used to determine if Ba ∆AT and Ba ∆CCAC-AT were significantly different, from which a slight degree of significance (p = 0.04) was concluded. However, in exploring this question using a Tukey test on various data sets, in no case was significance indicated. We therefore felt it was most appropriate to analyze a 3-way comparison with a Tukey test, and provide that as evidence that Ba ∆AT and Ba ∆CCAC-AT are not, in fact, significantly different.

• Fig 4C: One value had been entered wrong in producing the graph; this value has been corrected and is reflected in Fig 4C, S4D File, Fig 4 legend, and relevant text. This correction does not change the conclusion that Ba410 and Ba410∆CCAC are not significantly different.

 

Response to “Comments_230603”:

Major points:

1) Response: 

This issue is addressed in two places: 

In the Discussion/Transcription driven by minke whale 5’ flanking regions is only 8% that of humans/paragraph 5: We point out that mouse C2C12 have been widely used for many years, but discuss and reference work that supports the possibility that they may not be a perfect model system for whale studies, due to possible variations in transcription factor activity from other commonly used model systems and from the in vivo situation.

In the Discussion/Limitations of the study/paragraph 3: We again raise this issue, and note that the same criticism can be made for the human CCAC-box in that its definition was also done in non-human model systems. Repeating experiments designed to define the CCAC-box in human and (as yet unavailable) cetacean muscle-cell culture systems would be of interest. 

We also discuss mechanisms through which the activities of the minke and human CCAC-boxes may differ in the Discussion/The minke whale gene has substituted a G-rich sequence for the CCAC-box/paragraphs 7 and 9 (“enhancer grammar”, and possible binding of MYOD to the human but not minke CCAC-box). 

2) Response:

We have chosen to use an experimental sample with an activity very close to “1.0” for the statistical comparisons in each set (figure); in this way we avoid always using the same wild-type Ba710 data due to the statistical problems this would have caused. In each set (figure) there was always one or more samples that could represent the null hypothesis: that is, the mutation does not have an effect on expression. We appreciate that this can cause confusion when comparing different sets of experiments, but each figure is scaled to the activity of wild-type Ba710 = 1.0 by the data analysis, which is now described explicitly in S10 File. We felt this was the least problematic approach to the data. 

So, addressing point (a), each sample was normalized in two steps: firefly luciferase counts in each well was normalized to an internal renilla luciferase standard included in the transfection to control for well-to-well variation, and each plate was normalized to wild-type Ba710 = 1.0 to control for plate-to-plate variation. The result is that on the Y-axis in each graph the 1.0 value is that of the wild-type Ba710 construct in each particular transfection plate.

Regarding point (b), the Tukey test does compare each sample against each other sample, and a summary of the Tukey test is shown in the Supplemental data files. The authors felt that including all of the comparisons (e.g., A vs B, A vs C, A vs D, B vs C, B vs D, and C vs D) was confusing, and it would be clearest to only present the chosen control against each of the others. We have added a Supplemental file (S10 File) to address your concern with this point. It contains all the data to allow readers to test the various possible comparisons; the samples designated as “significantly different” are generally highly significant, and so are quite robust to this kind of playing around with the data. S10 File also contains an example of how raw luciferase counts were converted to the “F/R/Ba” number so the reader can better understand where the value of 1.0 comes from.

For point (c), the actual P-values are stated in the legend to each figure. The authors felt that showing asterisks on the graph itself rather than P-values would be clearer for the reader. 

Minor points:

3. We appreciate that the title was a bit awkward. Perhaps part of the problem was caused by mixing the technical term “cetacean” with the colloquial term “human”? We have changed it to “Comparative analysis of the myoglobin gene in whales and humans reveals evolutionary changes in regulatory elements and expression levels”. We hope this is better.

4. The manuscript has been carefully read and subjected to Word spell- and grammar-checks. We believe we have caught all the typos.

5. We have changed the “functional swap” comment in the Discussion/ The minke whale gene has substituted a G-rich sequence for the CCAC-box/paragraph 8 to “Sequence flexibility in these runs of G-C nucleotides may have facilitated the proposed evolutionary switch between the CCAC-box and the G-rich sequence.” We hope this clarifies the meaning.

6. We appreciate that our original Fig 7B was complicated, so we have simplified it to make it more accessible. A link in the legend will take the reader to the original at the Genome Browser site for those who are interested.

 

Response to “Comments to the authors”:

Major point:

The authors thank this reviewer for their insightful comments and suggestions of references that were not included. A fresh consideration of the work of Isogai et al has changed our perception of the significance of our own work, and we have changed our discussion accordingly (Discussion/Transcription driven by minke whale 5’ flanking regions is only 8% that of humans/paragraphs 7 and 8). We had previously focused on regulatory mechanisms operating in real time (increasing translation or protein stability), rather than the myoglobin protein evolution that has occurred. 

We briefly review in this section the evolution of stability and solubility of the MB protein and in this way better include the work of Isogai et al and Berenbink, and address the question of “why cetacean did take strategy to improve the protein stability (and solubility) rather than to improve the transcription rates”. We conclude that “Therefore, high transcription levels are not necessary to account for high MB protein levels in adult cetacean animals” and we note that “Evolving an increased rate of transcription to satisfy the physiological need for high MB levels in the absence of protein evolution would not have solved the aggregation problem. Evolving instead a stabilized protein resistant to aggregation is a strategy widely shared among diving mammals.”

Minor point:

The tissue samples we had at our disposal were primarily representative of the mysticetes due to the history of this work (Cartwright et al. 2016. Tracking the development of muscular myoglobin stores in mysticete calves. PLoS One, 11(1), p.e0145893). We did have dolphin and porpoise available but did not have sperm whale samples. 

Due to the consistently low levels of transcription from the seven species tested, as well as the related terrestrial species (cow, elk, pig), expanding our survey did not seem to be a priority. 

The point is well taken, however, and perhaps future work could include sperm whales and beaked whales (probably the diving world record holders). It would also be of interest to include Weddell seals in a direct comparison to whales, since seals are such a distant group of diving mammal.

 

Response to Reviewer #3:

The authors thank this reviewer for their suggestion of references to include, and we have done so. In the Discussion/Transcription driven by minke whale 5’ flanking regions is only 8% that of humans/paragraph 7 we point out that myoglobin levels increase steadily throughout the life of the animal, and reference these papers. In response to the linkage between low transcription levels and high adult protein levels, we conclude that, due to the evolved stability of the protein, high transcriptional rates are not required for the eventual accumulation of high protein levels.

The last two “minor points” are also relevant to this issue and the authors thank this reviewer for bringing them up as well. One of the other reviewers has pointed us to the work of Isogai et al and Berenbink regarding the evolution of the myoglobin protein toward greater solubility, more stable folding, and resistance to aggregation in diving mammals. We therefore propose that the evolutionary “reasoning” behind low transcription levels is that high transcription levels without this concomitant protein evolution would lead to myoglobin aggregation and destruction, and would therefore be counterproductive, as this reviewer has pointed out. Again, due to the increased stability of the protein, slow and steady does the job. 

In a similar vein, one may speculate that the evolution of prolonged maternal care and maternal coaching of energetic behaviors such as breaching may play into this issue. The induction of increased transcription by signaling mechanisms accompanying such behavior may be able to be titrated by the mom-coach for an optimal tradeoff between expression levels and protein accumulation. This is outside the scope of this particular paper, however.

We have also removed the implication that low transcription and protein stability “co-evolved”. The reviewer is absolutely correct in pointing out that the low transcription levels are most likely to be ancestral, and we raise this point in paragraph 8 of this section. It improves the discussion, and thank this reviewer for bringing it up.

Other minor points:

Thank you for the correction regarding capitalization of Cetacea and cetaceans. This has been corrected.

The sentence “Traditionally, evolution has focused on changes in amino acid sequence” has been changed to read: “Traditionally, the study of evolutionary change has focused on changes in amino acid sequence…” We hope this is clearer.

“the Cetacea” has been corrected to “baleen whales.

The manuscript has been carefully read for typos, and run through Word spell check. We believe we have corrected all typos.

---

## [Decision Letter · Decision Letter 1]

15 Aug 2023

Comparative analysis of the myoglobin gene in whales and humans reveals evolutionary changes in regulatory elements and expression levels

PONE-D-23-10581R1

Dear Dr. Sackerson,

We’re pleased to inform you that your manuscript has been judged scientifically suitable for publication and will be formally accepted for publication once it meets all outstanding technical requirements.

Kind regards,

Julie Dumonceaux

Academic Editor

PLOS ONE

Additional Editor Comments (optional):

Reviewers' comments:

Reviewer's Responses to Questions

**Comments to the Author**

1. If the authors have adequately addressed your comments raised in a previous round of review and you feel that this manuscript is now acceptable for publication, you may indicate that here to bypass the “Comments to the Author” section, enter your conflict of interest statement in the “Confidential to Editor” section, and submit your "Accept" recommendation.

Reviewer #3: All comments have been addressed

Reviewer #4: All comments have been addressed

2. Is the manuscript technically sound, and do the data support the conclusions?

Reviewer #3: Yes

Reviewer #4: Yes

3. Has the statistical analysis been performed appropriately and rigorously? 

Reviewer #3: Yes

Reviewer #4: Yes

4. Have the authors made all data underlying the findings in their manuscript fully available?

Reviewer #3: Yes

Reviewer #4: Yes

5. Is the manuscript presented in an intelligible fashion and written in standard English?

Reviewer #3: Yes

Reviewer #4: Yes

6. Review Comments to the Author

Reviewer #3: (No Response)

Reviewer #4: The authors have sufficiently responded to the comments of this reviewer, and this reviewer recommends this manuscript for publication.

7. PLOS authors have the option to publish the peer review history of their article (what does this mean?). If published, this will include your full peer review and any attached files.

Reviewer #3: **Yes: **Kevin L. Campbell

Reviewer #4: **Yes: **Tsuyoshi Shirai

---

## [Editor Report · Acceptance letter]

21 Aug 2023

PONE-D-23-10581R1 

Comparative analysis of the myoglobin gene in whales and humans reveals evolutionary changes in regulatory elements and expression levels 

Dear Dr. Sackerson:

I'm pleased to inform you that your manuscript has been deemed suitable for publication in PLOS ONE. Congratulations! Your manuscript is now with our production department. 

Kind regards, 

on behalf of

Dr. Julie Dumonceaux 

Academic Editor

PLOS ONE